# Reconciliation and evolution of *Penicillium rubens* genome-scale metabolic networks– What about specialised metabolism?

**Delphine Nègre**[1,2☯], **Abdelhalim Larhlimi**[2☯], **Samuel Bertrand**[1☯]*

**1** Nantes Université, Institut des Substances et Organismes de la Mer, ISOMer, Nantes, France, **2** Nantes Université, École Centrale Nantes, CNRS, Nantes, France

☯ These authors contributed equally to this work.
* samuel.bertrand@univ-nantes.fr

**Data Availability Statement:** All relevant data are within the paper and its Supporting Information files.

## Abstract

In recent years, genome sequencing of filamentous fungi has revealed a high proportion of specialised metabolites with growing pharmaceutical interest. However, detecting such metabolites through *in silico* genome analysis does not necessarily guarantee their expression under laboratory conditions. However, one plausible strategy for enabling their production lies in modifying the growth conditions. Devising a comprehensive experimental design testing in different culture environments is time-consuming and expensive. Therefore, using *in silico* modelling as a preliminary step, such as Genome-Scale Metabolic Network (GSMN), represents a promising approach to predicting and understanding the observed specialised metabolite production in a given organism. To address these questions, we reconstructed a new high-quality GSMN for the *Penicillium rubens* Wisconsin 54–1255 strain, a commonly used model organism. Our reconstruction, *i*Prub22, adheres to current convention standards and quality criteria, incorporating updated functional annotations, orthology searches with different GSMN templates, data from previous reconstructions, and manual curation steps targeting primary and specialised metabolites. With a MEMOTE score of 74% and a metabolic coverage of 45%, *i*Prub22 includes 5,192 unique metabolites interconnected by 5,919 reactions, of which 5,033 are supported by at least one genomic sequence. Of the metabolites present in *i*Prub22, 13% are categorised as belonging to specialised metabolism. While our high-quality GSMN provides a valuable resource for investigating known phenotypes expressed in *P. rubens*, our analysis identifies bottlenecks related, in particular, to the definition of what is a specialised metabolite, which requires consensus within the scientific community. It also points out the necessity of accessible, standardised and exhaustive databases of specialised metabolites. These questions must be addressed to fully unlock the potential of natural product production in *P. rubens* and other filamentous fungi. Our work represents a foundational step towards the objective of rationalising the production of natural products through GSMN modelling.

**Funding:** SB was funded by the French national research agency (ANR - Agence National pour la recherche - https://anr.fr/) for this project; project number ANR-18-CE43-0013. The funders had no role in study design, data collection and analysis, decision to publish, or preparation of the manuscript.

**Competing interests:** The authors have declared that no competing interests exist.

## Introduction

Comparable to Alexander Fleming's pioneering discovery of penicillin in 1929, the elucidation of natural products (NPs) has frequently been serendipitously achieved through the random screening of plant, fungal, and bacterial extracts. Natural molecules, synthesised by living organisms, can be categorised into two types based on their function within them: constitutive metabolites, which are essential for the organism's growth, and specialised metabolites, which provide competitive advantages towards the organism's environment, historically referred to as secondary metabolites. The latter is of great interest in pharmacology, both in the field of health and agriculture, since many drugs are originated from these biologically active natural molecules [1]. However, since the early 2000s, the discovery rate of new biologically active NPs has slowed down or even become scarce. Moreover, the time-consuming, inefficient and costly traditional methods used for bioprospecting new NPs have been slowly abandoned [2]. Given the increasing antibiotic resistance [3], it is essential to reinvigorate research on NPs, which has faltered in recent years, using more rational and predictive approaches. The confluence of genomics and metabolomics has now enabled the correlation of metabolites with their structural data to their respective biosynthetic gene clusters (BGCs). Combined with the automation and qualitative evolution of tools observed in recent years [4], these approaches have become increasingly appealing for new bioactive NPs discovery.

Filamentous fungi are known for their ability to produce a diverse array and a substantial amount of NPs [5]. Over the years, significant knowledge has been accumulated about their metabolism, and fungal genomes exploration has highlighted the high proportion of specialised metabolites produced by enzymes with co-located genes. In addition to biosynthetic capabilities, such BGCs could be associated with resistance, transport, or regulatory genes. Thus, their evolutionary history may provide clues to their biological activities that could guide molecular engineering research [6]. However, despite the richness of BGCs exhibited by fungi, one of the extensive risks NP chemists face is the strong tendency to rediscover previously characterised molecules [2]. Such a trend is inconsistent with the genomic data that have been acquired, showing a large number of gene clusters associated with unknown specialised metabolism. Indeed, the differences observed between theoretical *in silico* models and *in vitro* cultures indicate that gene clusters not expressed in the laboratory are numerous and form an untapped resource for novel bioactive NPs [7,8].

*Penicillium rubens*, formerly known as *Penicillium chrysogenum* [9], is a fungus used in the industry to produce *β*-lactam antibiotics, including penicillin. Consequently, both industrial and wild terrestrial strains of *Penicillium* have been extensively studied over the years. After decades of classical strain improvements using the wild strain NRRL1951 [10,11], all industrial strains with high penicillin yields are now derived from the Wisconsin 54–1255 strain, which has therefore become a model species within the filamentous fungi [12]. Thus, the availability of its genome [13] makes it a worthy candidate for studying its metabolism. In 2009, 34 BGCs were identified in *P. rubens*, only 4% of which were associated with a known and isolated compound. Nine years later, 28% of the compounds from the now-identified 50 BGCs were characterised [10]. Thus, understanding the conditions under which silent, orphan or cryptic gene clusters could be activated remains of great interest. Following the example of what has been achieved for bacterial NPs [14], decoding the genome is the crucial first step toward understanding how to unlock such cryptic genes.

In systems biology, an organism's biochemical and physiological metabolic properties can be described by Genome-Scale Metabolic Networks (GSMNs) studies [15]. Reconstruction of such models is based on computational methods using both *in silico* and experimental data. Indeed, integrative biology principles applied to genomics and metabolomics allow us

nowadays to link the high-quality data obtained by analytical technologies to the corresponding models. Moreover, increasing model complexity aims to integrate and visualise heterogeneous knowledge-related metabolism more and more efficiently [16], thereby enhancing our understanding of metabolism at the system level [17]. However, sustainability, evolution, and exploitation of models are intrinsically linked to their adoption by the scientific community, as illustrated by the work performed on the *Caenorhabditis elegans* metabolic network reconstruction [18]. Unfortunately, only a few examples of individual network reconciliation are reported [19,20], aside from the well-studied model organisms presented in a non-exhaustive list by Gu *et al.* [21], which are exceptions. This phenomenon can be attributed to various factors, including the massive generation of data associated with its quality and its updating, the lack of inter- and intra-operability between platforms and software, the multiplication of reconstruction tools linked to the current facility to reconstruct GSMNs, and the lack of transparency in the acquisition of previous networks [22–25]. Moreover, a GSMN represents a platform of organised and summarised resources intended to reflect the optimal metabolic capabilities of the target organism at the time of reconstruction based on the available knowledge and data at that moment (*e.g.* knowledge snapshot). In this respect, given the constant and rapid evolution of data, the need to maintain these models up-to-date and standardised is increasingly becoming pressing.

Therefore, since the first *P. rubens* GSMN (*i.e. i*AL1006) was published in 2013 [26], the work presented here proposes a new GSMN for *P. rubens* Wisconsin 54–1255, considering the points mentioned above. The present study is primarily concerned with the reconstruction process, with particular emphasis on providing informative outcomes regarding specific model generation. Additionally, particular attention was paid to specialised metabolism. Thus, the provided model could be compatible with future research on NPs pathways regulation, which could furnish insight into cryptic pathways. Thus, the *i*Prub22 model, composed of 5,919 reactions and 5,192 unique metabolites, combines data from previous reconstructions, an updated functional annotation, a homology search with various phylogenetically close and distant organisms, and manual curation based mainly on literature. Furthermore, our reconstruction provides the necessary framework for investigating the growth of *P. rubens* and exploring the production of its specialised metabolites under varying media conditions through constraint-based modelling.

## Results

### Draft generation: Reconstruction and reconciliation

The GSMN reconstruction *i*Prub22 started with the automatic generation of two drafts generated using genome functional annotation and homology searches with pre-existing models.

**Functional annotation subnetwork.** Usually, three annotation labels are required to provide a source of 'annotation-based' metabolic reactions. Thus, Enzyme Commission (EC) numbers, Gene ontology terms (GOT) [27,28], and protein domains (PFAM) were sought among the *P. rubens* sequences. Of the 12,556 gene sequences that constitute the *P. rubens* genome [13] 8,948 (71%) possess at least one annotation (*i.e.* EC number, GOT and/or PFAM), and 2,270 (25%) are annotated by all three annotations (S1 Fig in S1 File). The EC numbers are associated with genomic sequences based on KEGG orthology (KO) identifiers [29]. Thus, the combined use of KAAS [30] and gene information stored in KEGG leads to an 11% enrichment in EC numbers compared to relying solely on Trinotate [31] results (S1 Fig and S1 Table in S1 File). The specificity and complementarity of these three annotation sources are illustrated in S2.A Fig in S1 File. As a result, these annotations allowed the generation of

the functional annotation subnetwork composed of 3,390 genes (*i.e.* 27% of the genome), 1,359 pathways, 3,205 enzymatic reactions and 3,577 metabolites.

**Orthology subnetwork.**   In parallel, following the search for homology between species, a second subnetwork was reconstructed. Orthologous genes between *P. rubens* and seven other reconstruction templates were searched to achieve the creation of this subnetwork. Orthology templates were chosen according to the quality of their GSMNs, their phylogenetic proximity to *P. rubens*, or their reconstruction procedure (see the selected network's topology in S3 Fig in the S1 File).

Only the genomic sequences initially present in the GSMNs are queried using OrthoFinder [32]. Respectively, the templates *Arabidopsis thaliana* [33], *Aspergillus nidulans* [34,35], *Aspergillus niger* [34,35], *Neurospora crassa* [36], *Penicillium chrysogenum* species complex [34,35], *Saccharina japonica* [37] and *Schizosaccharomyces pombe* [34,35] are composed of 2,330, 1,279, 1,299, 836, 1,352, 5,016 and 810 sequences of which only 0.04%, 1.9%, 0.15%, 1.7%, 11%, 0.04% and, 0.49% were not found in the constitution of the queried proteomes. The 8,519 homologous sequences between *P. rubens* and the different templates *A. thaliana*, *A. nidulans*, *A. niger*, *N. crassa*, *P. rubens* species complex, *S. japonica*, and *S. pombe* are distributed according to 478, 942, 866, 635, 1,003, 1,273 and, 667 orthogroups respectively (S2 Table in S1 File and S2 File). Sequence distribution among the different species templates and orthogroups is displayed in S4 Fig in the S1 File. Of these homologous sequences, 2,903 are paralogs and will not be considered for reconstruction. Respectively, there are 1,494, 1,144, 1,049, 766, 1,174, 2,302, and 764 orthologs between the templates studied (*i.e. A. thaliana*, *A. nidulans*, *A. niger*, *N. crassa*, *P. rubens* species complex, *S. japonica*, and *S. pombe*) and *P. rubens*. Complementarily, the number of *P. rubens* genes being orthologous to the different templates are 866, 1,818, 1,620, 1,104, 2,368, 2,329, and 1,114, respectively.

Orthologous genes to the template ones were used to obtain their corresponding reactions related to *P. rubens* Wisconsin 54–1255. As the selected template networks were mostly reconstructed between 2010 and 2018 *via* the KEGG database [29], a mapping step on the MetaCyc database [37] was necessary for the interoperability between the OrthoFinder results and the subnetworks inference. The draft generation indicates that less than 50% of the detected and potentially informative reactions were lost during this mapping (S2.B Fig in the S1 File). Mapped reactions correspond to 46% of the total reactions for *A. thaliana*, 49% for *A. nidulans*, 49% for *A. niger*, 49% for *P. rubens* species complex, 49% for *S. pombe* and 46% for *N. crassa*. The best mapping rate (87%) is obtained with the *S. japonica* network, whose GSMN was reconstructed following a similar protocol to the one presented here and directly *via* the MetaCyc database [37]. In terms of gene number, these losses represent 18% of the total number of genes detected by OrthoFinder (*i.e.* 5,616 genes). However, 70% of the genes (*i.e.* 716 genes) were finally included in the GSMN as they possess correspondence *via* the annotation or with external sources (S5 Fig in S1 File). The remaining unmapped 307 sequences were lost and represent a set to be preferentially explored for future network improvements. Thus, the orthology subnetwork is composed of 4,615 genes, 1,550 pathways, 2,946 enzymatic reactions and 3,238 metabolites.

**Enrichment with external sources.**   *Penicillium rubens* is a widely studied model organism, but the available data heterogeneity makes conciliation a delicate task. Therefore, the choice to add external data was made on consistent sources, namely PchCyc, the Pathway/Genome Database (PGDB) from BioCyc [38], and the latest version of the organism's network, called Prubens [39]. This reconstruction, resulting from a large-scale automatic reconstruction process, takes advantage of a mapping on MetaCyc from the original version of the *i*AL1006 network [26]. Thus, data were added when reactions, or in default, all their metabolites, had compatible MetaCyc identifiers and were supported by at least one Gene-Protein-Reaction

(GPR) association. Respectively, Prubens and PchCyc are composed of 2,533 and 1,291 reactions, of which 650 are shared. Of the 3,824 total reactions queried, 2,418 had a compatible MetaCyc identifier, and 2,112 were associated with at least one gene. Furthermore, of the 1,725 reactions already present in the draft, 313 had identical GPRs associations, 123 had no genomic information in the external sources (*e.g.* in the *i*Prub22 draft, these reactions are henceforth annotated between 1 and 39 genes), and 1,289 possessed different GPR associations. Finally, the draft was completed by 510 reactions (*i.e.* 440 reactions belonging exclusively to Prubens, 43 reactions belonging exclusively to PchCyc and 27 reactions coming from both sources). Moreover, 263 supplementary reactions from *i*AL1006 were also included since all the reactants/products of those reactions possess a MetaCyc identifier. To ensure data traceability, the original identifiers of these reactions were kept, and the selection is shown in S6 Fig in the S1 File.

**The initial draft.**   All the data obtained were then merged to produce a first version of the network (*i.e.* draft). The complementarity of each approach, displayed in Fig 1 and Supporting Information (S7 S2.C, and S8 Figs in the S1 File), makes it possible to question the meaning and relevance of the added data. These figures trace the origin of each reaction and, therefore, of the genes and metabolites, within the draft. Thus, it is easier to disentangle the information from biological reality or computational bias. For instance, in the orthologous subnetwork, 17 and 1,237 reactions (S5.C Fig in the S1 File) were exclusively from the most phylogenetically distant templates (*i.e. A. thaliana* and *S. japonica*, respectively). In the case of the 17 *A. thaliana* reactions, 8 (47%) of them are also supported by the annotation source, and for *S. japonica*, the number amounts to 967 (78%) reactions. Similarly, the addition of external sources (*i.e. i*AL1006, Prubens and PchCyc) supported 773 reactions (17% of the draft reactions) which were not retrieved with the annotation and orthology subnetworks. Therefore, reactions supported only by one source should be given special attention in the future.

Moreover, according to the nomenclature of EC numbers, reactions are divided into the following eight categories: oxidoreduction 28%, transfers reactions 24%, hydrolyses 18%, lyses 7.9%, isomerisation 2.8%, reactions involving ligases 3.4%, translocation 0.38% and undetermined reaction type 16%. Nevertheless, this distribution is different within the genes since the most frequent classes are: transferases (29%), hydrolases (26%) and oxidoreductases (24%). However, it should be noted that only 42% of the genes have a clearly identified EC number, predominantly as a functional annotation result (S9 Fig in the S1 File).

Furthermore, from the 4,959 metabolites present at this reconstruction stage, 1,085 (22%) come exclusively from the annotation's subnetwork, 772 (16%) from the orthology subnetwork and 565 (11%) from external sources (Fig 1C). Consequently, 2,537 (51%) metabolites are shared between at least two different sources (S10 Fig in the S1 File). According to the MetaCyc compounds' ontologies, 349 (*i.e.* 7% of draft metabolites) belong to the specialised metabolism (*i.e.* 41 (12%) exclusively from the annotation, 92 (26%) exclusively from the orthology, 153 (44%) exclusively from external sources and the remaining 63 (18%) are supported by at least two of these sources). MetaCyc v23.0 contains approximately 12% of compounds annotated "secondary".

*In fine*, the reconstructed draft includes 5,585 genes which support 4,916 reactions included in 1,817 pathways and involving 4,959 metabolites.

## Reconstruction improvements

Once the initial draft is produced, several modifications must be performed to obtain a functional and searchable model compatible with the experimental observations (*i.e.* a knowledge platform comprising a set of accessible resources for a given organism at the instance of

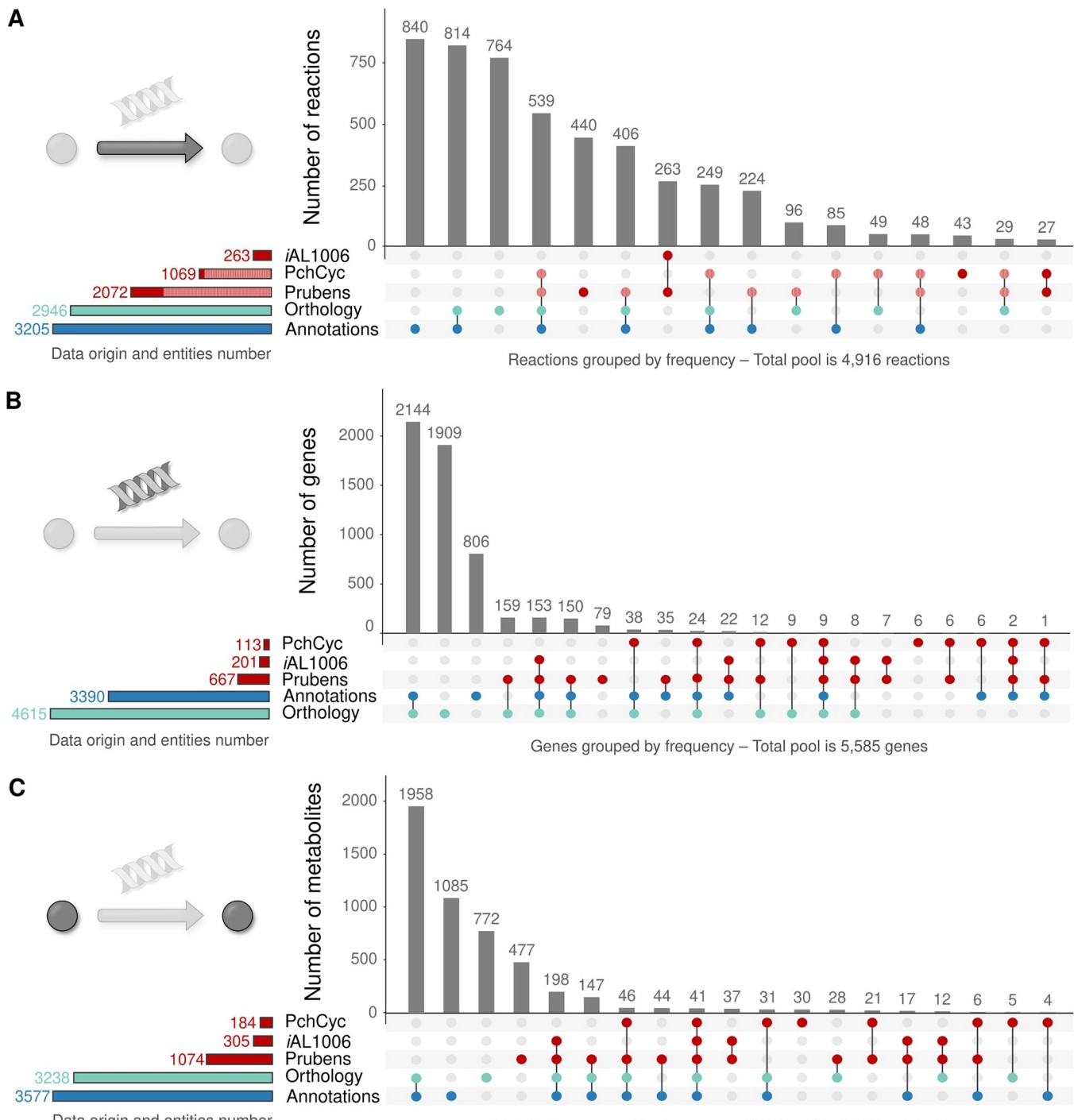

**Fig 1.** UpSet diagram representing the origin of each of the reactions (A), genes (B), and metabolites (C) added to the *Penicillium rubens* Wisconsin 54–1255 draft. This traceability underlines the complementarity of the approaches. Information comes from annotation (🟦) orthology (🟩) subnetworks and external sources (🟥). The red hatched area (▦) corresponds to reactions shared between the external sources and the draft from the annotation and/or orthology subnetworks. For a further view and details, see S7 and S8 Figs in S1 File.

reconstruction). The first step is to establish a list of metabolites known to be produced by the studied organism. Initially, we investigated the presence of these target compounds in the draft and subsequently explored their topological producibility. When these compounds are either absent or not topologically producible, gap-filling steps should be performed to improve network connectivity. Therefore, the following section deals with the necessary prerequisites for setting up a high-quality GSMN ready for subsequent flux analyses.

**Target selection for model enhancement.** As it is not possible to bring the same confidence weight to all available data, the targets selected to study and improve the metabolic network of *P. rubens* were grouped into three distinct sets of targets (*i.e.* set 'Targets1' with high confidence, set 'Targets2' with medium confidence and set 'Targets3' with low confidence–S3 File for further details).

The first target set corresponds to information from the literature. The target metabolites belong indifferently to specialised or constitutive metabolism in that particular case. Two publications, in particular, caught our attention, one allowing the accurate modelling of the sugar pathway in fungi [40] and the other resulting from the penicillin biosynthesis pathway manual curation [41]. This list was completed by metabolites related to the biomass function [26]. In addition, the SMASH tool suite [42] was used to highlight 19 specialised metabolite pathways through gene cluster search. Although, only the biosynthetic pathways of penicillins, roquefortines and patulin can be exploited due to the lack of MetaCyc identifiers for compounds of the other pathways. However, knowing that in *P. rubens*, the patulin biosynthetic pathway is incomplete [7], this compound is discarded from 'Targets1', and its pathway curation was later performed in this sense. Consequently, 11 metabolites were supported by genome mining sources in the first list of targets to be monitored. During this process search, a list of 46 compounds lacking MetaCyc identifiers was also generated (*i.e.* orphan metabolites). Moreover, even if Melanin and PR-Toxin are not strictly speaking orphan metabolites since they are referenced on MetaCyc, they cannot be produced topologically since no reactions were associated with them. At that stage, those latter compounds cannot be monitored within the network unless focusing on their closest referenced precursor. Finally, set 'Targets1' is thus composed of 243 targets.

The second list is obtained by querying specific NP databases. Using a taxonomy-oriented query from the naturaL prOducTs occUrrence databaSe (LOTUS) [43], 240 compounds for *P. chrysogenum*/*P. rubens* are retrieved along with their InChIKeys, which are used to search within the MetaCyc database. Based on an exact InChIKey match, followed by an analogue of the first two sections of the identifier and finally on the first part (*i.e.* corresponding to planar chemical structure search), only 8, 11 and 47 metabolites were found, respectively. Depending on these criteria, between 4, 7 or 13 of these metabolites were already present in the draft. Therefore, in the best case, only 20% of the information in LOTUS can be included in the reconstruction. The assignment verification to the studied strain allows the selection of 35 metabolites which constitutes the set 'Targets2'.

Finally, the third set of targets was established based on the metabolites present in the reconstruction (*i.e.* at the step intermediate C–Table 1) and that have annotation with the term 'Secondary-Metabolites' in the MetaCyc compounds ontologies when they were searched with SmartTables. At that stage, the reconstruction contains 400 metabolites classified as "secondary". They are divided into 149 compounds that are at least produced and consumed by a reaction, 90 that are only consumed and 161 that are only produced (*i.e.* 251 topological dead-ends). Thus, Set 'Targets3' was composed of 400 targets.

Altogether, 653 metabolites were used to obtain a high-quality GSMN by seeking the network for their presence and connectivity. Based on the MetaCyc annotations, BGCs mining and LOTUS database, this list is divided into 215 and 438 compounds belonging to constitutive and specialised metabolism, respectively (Fig 2).

**Table 1. Potential topological producibility expressed in the different curation steps of the reconstruction.**

| | Reconstruction progress | | | | | | |
|---|---|---|---|---|---|---|---|
| | **Draft** | **A** | **B** | **C** | **D** | **E** | **GSMN** |
| **Number of entities** | --●-- | --●-- | --●-- | --●-- | --●-- | --●--→ | |
| **Reactions** | 4,916 | 5,389 | 5,475 | 5,536 | 5,551 | 5,823 | 5,919 |
| **Active reactions** | - | 2,685 (50%) | 2,775 (51%) | 2,953 (53%) | 2,970 (54%) | 3,527 (61%) | 3,707 (63%) |
| **Metabolites** | 5,060 | 5,245 | 5,292 | 5,309 | 5,318 | 5,437 | 5,464 |
| **Producible metabolites** | - | 1,700 (32%) | 1,760 (33%) | 1,887 (36%) | 1,899 (36%) | 2,406 (44%) | 2,545 (47%) |
| **Metabolites only consumed** | 1,564 (31%) | 1,527 (29%) | 1,507 (28%) | 1,484 (28%) | 1,485 (28%) | 1,366 (25%) | 1,333 (24%) |
| **Metabolites only produced** | 1,691 (33%) | 1,736 (33%) | 1,726 (33%) | 1,721 (32%) | 1,725 (32%) | 1,661 (31%) | 1,641 (30%) |
| **Metabolites consumed and produced** | 1,805 (36%) | 1,982 (38%) | 2,059 (39%) | 2,104 (40%) | 2,108 (40%) | 2,410 (44%) | 2,490 (46%) |
| **Producible metabolites (targets 1)** | - | 193/235 (81%) | 198/235 (84%) | 226/235 (95%) | 226/235 (95%) | 226/235 (95%) | 234/243 (96%) ▲ |
| **Producible metabolites (targets 2)** | - | 5/35 (14%) | 6/35 (17%) | 6/35 (17%) | 14/35 (40%) | 14/35 (40%) | 14/35 (40%) |
| **Producible metabolites (targets 3)** | - | 87/400 (22%) | 95/400 (24%) | 147/400 (37%) | 147/400 (37%) | 368/400 (92%) | 368/400 (92%) |

The main objective of gap-filling was to ensure that the known metabolites in *Penicillium rubens* were present before using the reconstruction as a model for analysis. As gap-filling is an iterative process, it was linearised, and results were grouped for clarity's sake. Thus, between the draft and the GSMN, intermediates A, B, C, D and E were generated. First of all, to test the metabolites' producibility, 208 transport, 37 demand, 1 sink and 217 exchange reactions were incorporated (draft to A), followed by 86 spontaneous reactions (A to B) and then 61 (B to C), 15 (C to D), 272 (D to E) and 75 (E to GSMN) reactions from the four gap-filling runs. Minor corrections (not described) were made between E and the final reconstruction. The topology analyses were carried out with the seed list defined from the metabolites present in the extracellular medium. The eight metabolites specific to the biomass reaction were only monitored from their inclusion in the reconstruction at the GSMN step (▲).

**Modelling exchanges with the environment.**   To check the producibility of a compound by topological analysis and then by constraint analysis, it is necessary to define a set of initiation metabolites called seeds. Compounds selection from the extracellular medium led us to review and modify the reactions related to the metabolite transport phenomena between the various model compartments.

Although the proposed model does not include intracellular compartmentation, initially, 76 reactions, supported by a total of 204 genes, involve transport mechanisms. The location of their gene products corresponds to the cell membrane for 107 of them. The 22 reactions that do not possess one of these genes in their GPR association were either corrected or blocked in the model (*i.e.* reactions that biologically model an intracellular transfer whereas it is extracellular in the reconstruction). The performed curation allowed adding 208 metabolite transfer reactions between the extracellular and intracellular compartments. At least one gene, for a total of 220, is associated with 91 of them. For modelling purposes, 37 demand reactions (*i.e.* irreversible transfer from the external to the intracellular environment) and 1 sink reaction were also included.

Then, exchange reactions corresponding to 185 uptake reactions and 42 production reactions were added (S11 Fig in the S1 File and S4 File). Such exchange reactions are related to only the transfer of the same metabolites between the extracellular compartment and the environment (*i.e.* system boundaries). These reactions are entirely artificial. However, their addition will allow the simulation of the nutritional conditions of the organism (*i.e.* for further use of the model, *e.g.* tests of functionalities and production of metabolites of interest). Thus, the seeds list used for the topological analyses corresponds to the metabolites present in system boundaries (*i.e.* having an uptake reaction).

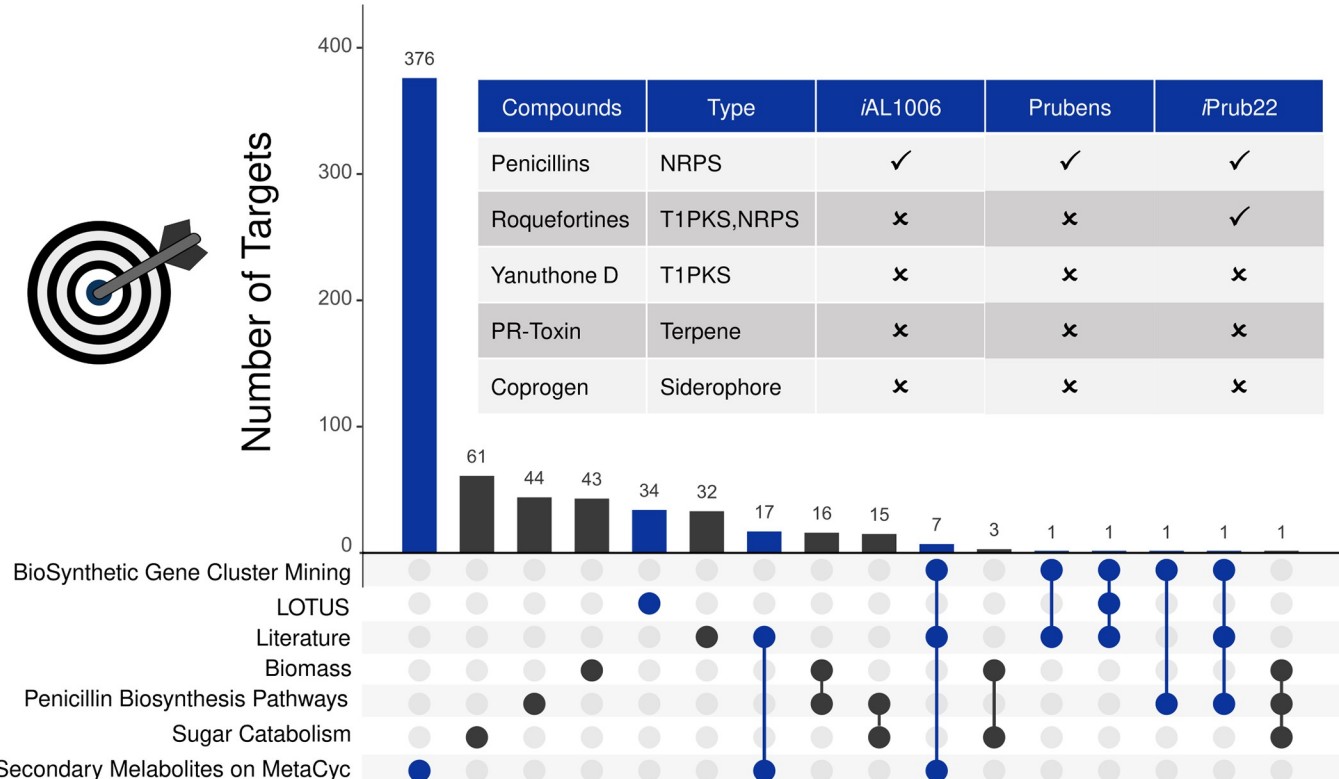

**Fig 2. Origin of the 653 targets used for reconstruction enhancement.** Example of secondary metabolites produced by *Penicillium rubens* according to the main natural product classes: NRPS (Non-Ribosomal Peptides Synthetase and siderophores), T1PKS (Type I Polyketide synthase), hybrid form, and terpenes. The selection was based on data from the literature (*i.e.* Biomass [26], Sugar Catabolism [40], Penicillin Biosynthesis Pathways [41] and literature details on Supporting Information), genome mining [42] and LOTUS databases [43]. All targets are divided according to constitutive (■ – 215) and specialised (■ – 438) metabolism. Metabolites found (✓) or not (✗) in the different versions of GSMNs are shown. Metabolites without MetaCyc-compatible information cannot be included in the networks (*i.e.* absence of compound identifier as for yanuthone D or the absence of reactions leading to compound production/degradation as for coprogen or PR-Toxin).

Finally, the reconstruction is composed of 284 (4.8%) transport reactions, 37 (0.63%) demand reactions, 1 sink reaction, 185 (3.1%) uptake reactions and 42 (0.71%) production reactions. These reactions, therefore, represent 9.3% of the GSMN reactions.

**Gap-filling and topological producibility.** Once the targets and seeds are defined, all that remains to be performed is to select available reactions for gap-filling querying. The database used for gap-filling (MetaCyc v23.0) contains 17,201 reactions. As gap-filling is a heuristic approach, these reactions are filtered into two categories to select the more relevant set of reactions to query. The filtering presented below reduces the space to be interrogated by 46%. Consequently, Gap-filling was run firstly with MetaCyc subset reactions (*i.e.* 9,281 reactions) with a high confidence weight.

The first category corresponds to reactions related to the fungi kingdom. This list was constructed based on a meta-network obtained by the reconstructions of a species' repertoires belonging to the fungi kingdom [44]. This meta-network is composed of 3,810 reactions. Interestingly, 77% of the reactions (*i.e.* 2,964 reactions) were already present in the draft. This list of reactions was completed by the BioCyc PGDBs of *Candida albicans*, *Exophiala dermatitidis* and *Saccharomyces cerevisiae*, composed of 1,320, 1,714 and 1,562 reactions, respectively. Combining these three latter networks yield a set of 2,905 reactions, among which 70% (*i.e.*

2,041 reactions) were already present in the draft. The fusion of these two lists of reactions provided a pool of 4,709 reactions, of which 64% (*i.e.* 3,036 reactions) were already included in the draft reconstruction. Finally, this first category comprises 1,043 reactions of interest.

At this point in the reconstruction, the draft network contains only reactions supported by at least one gene and thus is composed only of reactions catalysed by enzymatic proteins. For instance, biological catalysis carried out by RNA molecules (*e.g.* intra- or inter-molecular catalysis) or spontaneous reactions are not initially present in the reconstruction. However, their importance in modelling biological systems or under experimental conditions is no longer in question [45–47]. Therefore, the second category includes reactions for which the genomic information of the enzyme is absent or unknown, always according to data on MetaCyc. They were gathered by filtering the reactions on the MetaCyc web server into "Spontaneous reactions for which no enzyme is required" and "Enzyme-catalysed reactions for which no enzyme has been identified (pathway holes)". It allowed the selection of 705 and 4,971 reactions (downloaded in October 2021 and May 2022, respectively). However, it is noteworthy that 22 (3.1%) and 631 (13%) reactions belonging to these two sets, respectively, are already present in the network, thus questioning the relevance and accuracy of the annotations.

For the classification and traceability of the results, the process leading to the generation of the high-quality GSMN involved five intermediate steps labelled A to E. The initial step (*i.e.* A to B) combined the draft, which incorporated all data supported by at least one gene, with exchange reactions. In this step, 32% of the metabolites (*i.e.* 1,760 compounds) were topologically producible, and 51% of the reactions (*i.e.* 2,775 reactions) were active under the evaluated conditions. Through subsequent manual curation, these percentages were improved at the final step to 47% (*i.e.* 2,545 compounds) and 63% (*i.e.* 3,707 reactions), respectively. More particularly, out of the 645 monitored metabolites, 285 (43%) were found to be producible in the first intermediate step. In the final GSMN *i*Prub22, this percentage reached 94%. Indeed, the refinement process, summarised in Table 1, involved several gap-filling steps to enhance topological connectivity. For the first set of targets (*i.e.* 'Targets1'), 61 reactions carefully checked were incorporated into the reconstruction (*i.e.* B to C). Moreover, GPRs associations were included for 20 of them. For the second set of targets (*i.e.* 'Targets2'), 29 out of 35 metabolites were not producible after the first gap-filling run. Through the addition of 15 reactions from the second gap-filling (*i.e.* C to D), 8 of the previously not producible metabolites were successfully recovered. The third set of targets (*i.e.* 'Targets3') was used to improve network connectivity by targeting secondary metabolites initially present in the draft. The addition of 272 reactions (*i.e.* D to E) in the third gap-filling expanded the model capabilities since 507 supplemental metabolites could henceforth be produced, and 557 more reactions were activated. A final gap-filling step was performed with all the metabolites present in the initial draft against reactions with unknown enzyme sequences. It resulted in the addition of 75 more reactions (*i.e.* E to GSMN). In summary, 510 reactions were included in *i*Prub22 through the gap-filling processes (S5 File). About half of these reactions were sourced directly from the MetaCyc pre-filtered set (S12 Fig in the S1 File). Furthermore, there was an enrichment of 86 specialised metabolites, as the draft contained 405 metabolites labelled as "secondary metabolites" compared to 491 in the final GSMN.

**Reconstruction transformation into a constrained model.** From a reconstruction perspective, achieving a consistent flux model involves addressing common and known issues to prevent inconsistencies and inaccuracies in the model's flux behaviour. Therefore, the *i*Prub22 model underwent refinement through adjustments, such as determining appropriate reaction reversibility, resolving redundant reactions or identifying and eliminating infeasible cycles.

Firstly, during the reconstruction loading, it was discovered that 61 reactions contained errors due to the ambiguous compounds representation serving as both reactants and products

(*i.e.* stoichiometry represented by the metabolite name). The lack of clarity in their definition, attributed to the utilisation of metabolites belonging to classes or superclasses, led to 56 blocked reactions (*i.e.* reactions involving overly generic and unbalanced compounds) and the correction of 5 others. Furthermore, the intracellular compartmentation is not explicitly modelled within *i*Prub22. Consequently, 11 reactions related to the transport between the cytoplasm and organelles (*i.e.* mitochondria or peroxisomes) are inadequately annotated. As these reactions are not relevant to the current objectives of the model, they have been blocked.

Secondly, we focus on the significance of the reversibility of reactions. When the reaction directionality is unknown, it is often assumed as reversible, which can introduce biases in model development and comprehension. To address this, we utilised evolving database annotations and improved knowledge in the field. Thus, we subjected the 967 reversible reactions of the initial reconstruction to further scrutiny using the SmartTable tool on the MetaCyc server. Among the 787 reactions retrieved from MetaCyc, 98 had their reversibility updated by closing one of their bounds. In addition, 165 reactions were without annotation labels, indicating unknown directionality, and thus were blocked. These initial modifications yielded a model that exhibits sensitivity to nitrogen variations. Specifically, by manipulating the openings of various nitrogen sources, we discern changes in the flux distribution within the model. This sensitivity highlights the significance of nitrogen availability and its impact on metabolic processes.

Thirdly, we dealt with duplicate reactions contained within the reconstruction. As a result, we identified 49 pairs of similar reactions, with one possessing a MetaCyc identifier and the other using a "homemade" identifier due to the reconciliation process. In such a case, we retained the more balanced reaction from each pair. Furthermore, we meticulously examined the GPRs association, ensuring either their complete identity or the inclusion of all genes from one reaction into the other. Then, to maintain consistency within the stoichiometric matrix and ensure the reaction uniqueness, we identified and blocked 79 instances of duplicated reactions.

Fourthly, we encountered issues related to energy production when the model was fully closed, as well as complications with metabolite production. These observations suggested the presence of infeasible cycles within the network, which can disrupt proper metabolic functioning. Furthermore, at this stage, the model did not exhibit sensitivity to variations in carbon availability and sources, highlighting the need for further adjustments. To address these challenges, we blocked some of the non-equilibrated reactions. This step ensures the overall mass and charge balance of the model, guaranteeing that it adheres to fundamental principles of stoichiometry and improves its accuracy and reliability for subsequent analyses. Out of the 5,919 reactions present in the reconstruction, 1,398 (23%) were found to be mass imbalanced, indicating inconsistencies in the stoichiometry of the participating metabolites. Additionally, 969 reactions (16%) exhibited charge imbalance, implying a discrepancy in the overall charge of the reactants and products. Regarding the 5,464 metabolites encompassed in the reconstruction, 3,517 (65%) were exclusively involved in balanced reactions, demonstrating that their participation maintained the system equilibrium. Conversely, a small fraction of 192 metabolites (3.3%) lacked molecular formulae, emphasising the need for further annotation and characterisation. Among the 1,402 imbalanced reactions identified, 228 were exchange reactions, 10 were associated with biomass production and its assimilated reactions, 170 had been closed during previous curations mentioned earlier, 25 were indispensable for biomass production, 23 were involved in the synthesis of specialised metabolites, and 946 were effectively blocked. This intervention helped resolve the problems related to energy production and metabolite synthesis, leading to a more functional and realistic network representation. Indeed, this approach prevented the occurrence of energy production from non-physiological sources

when the model is closed, ensuring the model's compliance with fundamental principles of energy conservation and maintaining its biological realism.

Thus, once all these modifications have been applied (*i.e.* 1,389 reactions, 23% of the reactions in the reconstruction had at least one of their bounds modified), and when all uptakes are unrestricted, the resultant model consists of 1,667 active reactions, accounting for approximately 28% of the overall reconstruction. Notably, 392 (24%) of these reactions are reversible, indicating the system's capability to accommodate bidirectional fluxes. This proportion underscores the importance of considering the flexibility and adaptability of metabolic pathways in response to changing conditions. All of the above model modifications and information are reported in Supporting Information (S7 File) to be compliant with the MIASE standards [48].

Finally, the model quality was assessed based on its ability to simulate organism growth. Biologically, the biomass reaction models the organism synthesising essential amino acids, membrane lipids, and sugars. When the predicted growth rate is null, this implies the incompleteness of the GSMNs. It results either from missing reactions in the biosynthetic pathway or an accumulation of one or more reaction products due to the absence of a degradation reaction of these metabolites. Thus, to overcome this issue, manual analysis was carried out either by adding outward transport reactions or determining the missing reactions through similar organisms' GSMNs analysis. According to previous studies, *P. rubens* has a maximum growth rate of around 0.17 h$^{-1}$ when it grows on sucrose [49]. Wild-type *Penicillium* species were reported to grow on glucose/sucrose at a rate between 0.14 and 0.22 h$^{-1}$ [50]. In the *i*Prub22 model, we conducted simulations by setting import rates of sulphur, riboflavin, thiamine, phosphate, and iron to 10 mmol.gDW$^{-1}$.h$^{-1}$, while oxygen import is unlimited. The carbon source is supplied at a rate of 15 mmol.gDW$^{-1}$.h$^{-1}$, and the nitrogen source at 5 mmol.gDW$^{-1}$.h$^{-1}$. When amino acids serve as both carbon and nitrogen sources, their import rate is adjusted to 15 mmol.gDW$^{-1}$.h$^{-1}$. All other uptake reactions are closed. In natural conditions, *P. rubens* requires a nitrogen source for growth, and our model accurately reflects this dependency. Moreover, the inability of the fungus to grow when cysteine is used as a nitrogen source aligns with characteristics expressed by *P. rubens* in real-life conditions. Under the reference conditions of using glucose and ammonia as carbon and nitrogen sources, the model predicts a theoretical growth rate of 0.2944 mmol.gDW$^{-1}$.h$^{-1}$. Reasonably, a potential correlation is observed between fungal growth and the increasing carbon availability in the environment, as depicted by the variations in the carbon source represented in Fig 3. However, when we substituted glucose with sucrose as the carbon source, a threefold increase in production was observed compared to the theoretical value [49]. Furthermore, Fig 3 illustrates additional simulations conducted to evaluate the growth capacity of the GSMN model under various media conditions.

## Comparisons and evolution of *Penicillium rubens* GSMN

**Connectivity and metabolic coverage.** The GSMNs, *i*AL1006, Prubens and our reconstruction encompass 1,660, 2,574 and 5,919 reactions that connect 2,429, 3,058 and 5,464 metabolites, respectively. These networks are supported by 1,006, 1,786, and 5,703 different genes, bringing the number of *P. rubens* genes into *i*Prub22 (*i.e.* metabolic coverage) to 45%, surpassing the respective percentages of 8% and 14% observed in the first and second reconstructions from 2013 and 2018. Therefore, besides an increasing number of reactions associated with better metabolic coverage links to recent improvements in databases and annotation methods, a significant evolution observed over the years is logically the enhanced connectivity within the models themselves (Fig 4).

Among the 5,703 genes in *i*Prub22, 30% (3,771 genes) are newly added to the reconstruction, while 6% (800 genes) are shared with both previous network versions. Additionally, 7%

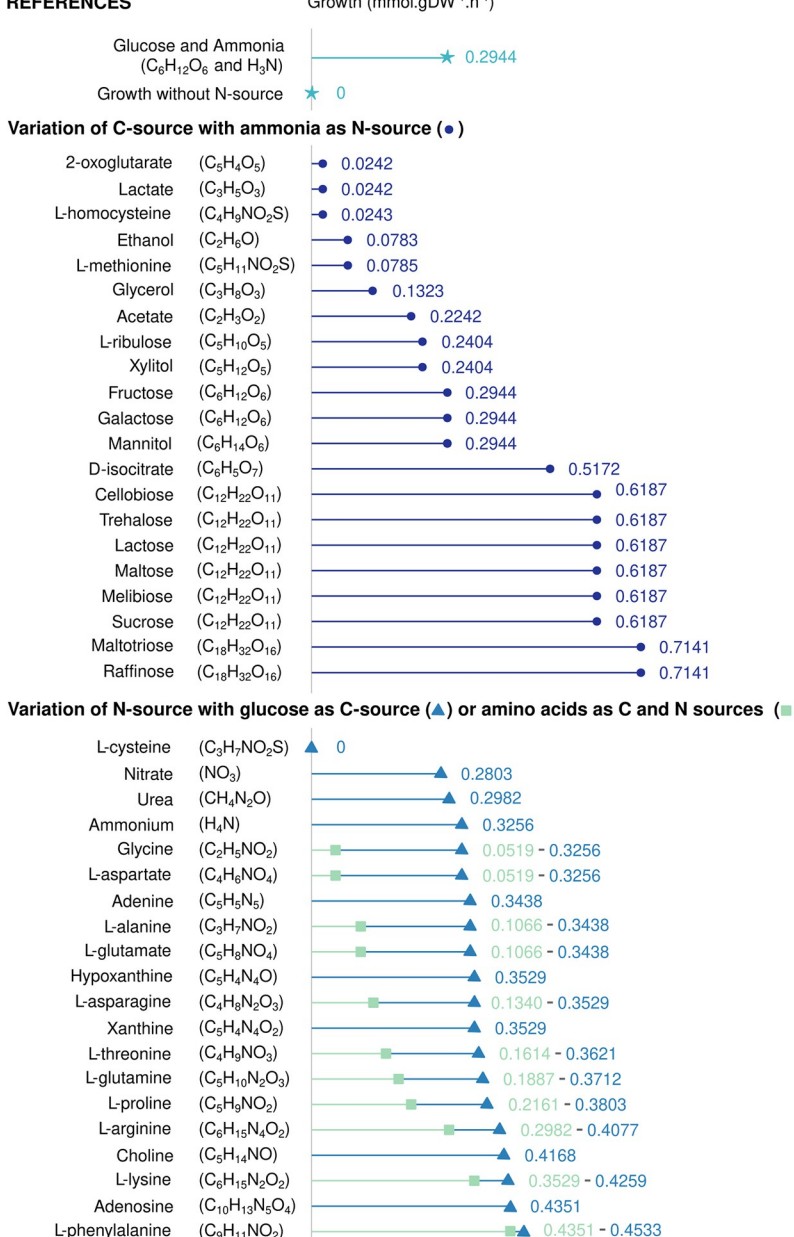

**Fig 3. Theoretical growth of *P. rubens* on different simulated media.** Reference conditions are denoted by the light blue star in the figure (★). The variations in the carbon source are represented by dark blue circles (●). The blue triangles (▲) represent the optimum biomass flux observed when the nitrogen source is altered. The green squares (■) indicate the maximum biomass flux achieved when an amino acid functions as carbon and nitrogen sources. Import rates of sulphur, riboflavin, thiamine, phosphate, and iron are set to 10 mmol.gDW$^{-1}$.h$^{-1}$, while oxygen import is unlimited. The carbon source is supplied at a rate of 15 mmol.gDW$^{-1}$.h$^{-1}$, and the nitrogen source at 5 mmol.gDW$^{-1}$.h$^{-1}$. When amino acids serve as both carbon and nitrogen sources, their import rate is adjusted to 15 mmol.gDW$^{-1}$.h$^{-1}$. All other uptake reactions are closed. For more detailed information, please refer to the additional data presented in the S6 File.

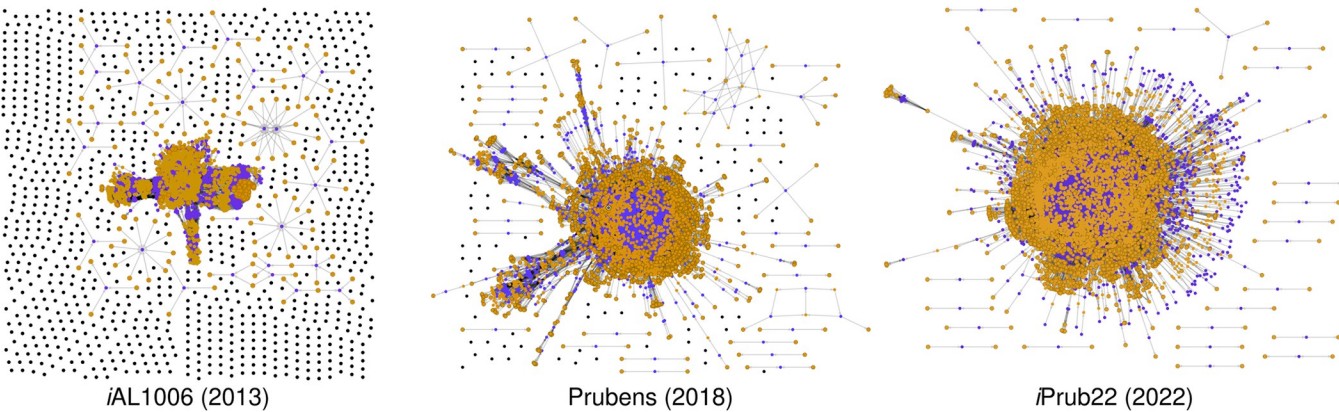

**Fig 4. *Penicillium rubens* network visualisation through bipartite graphs.** Illustration of the progressive evolution and improved topology of the *Penicillium rubens* network over time. The purple nodes (■) correspond to reactions, while the orange nodes (■) represent metabolites. Disconnected metabolites are depicted as black nodes (■). These images were generated using ModelExplorer (version 2.1) [51].

(939 genes) are shared with the most recent version of the network, Prubens, while 2% (193 genes) are mutual with the oldest, *i*AL1006. The 53 genes contained in previous versions and not found in *i*Prub22 are mainly involved in metabolite transport (data obtained with Fungi-Fun, not shown). According to the GOT annotation provided by FungiFun (S13 Fig in S1 File), the genes contributing to this new network have annotations covering various biological and metabolic processes such as antibiotic responses, signal transduction, protein phosphorylation and molecular binding. Also, the KEGG annotation indicates that processes involving DNA and RNA (*e.g.* nucleotide metabolism, transcription, translation, replication and repair) are present. These categories are reflected in the annotation provided by FunCat. The 3,771 genes included in the reconstruction are associated with a range of 1 to 61 reactions. On average, each gene is linked to approximately 4.5 reactions, with a median value of 2 reactions. Furthermore, among the 6,884 sequences that are not included in the *i*Prub22 reconstruction, approximately 61% of them have a substantial length capable of encoding functional enzymes, with an average size of approximately 370 amino acids (Fig 5).

**Specialised metabolism through versions.** The initial GSMNs, *i*AL1006 and Prubens, involved 1,395 and 3,058 metabolites distributed across 5 and 14 compartments, respectively. The evolution and comparison of *i*Prub22 metabolites with those of previous networks are based on MetaCyc (v23.0) identifiers for Prubens and the first section of InChIKeys generated from InChI for *i*AL1006. Indeed, due to the identifiers incompatibility, the comparison with the 2013 network was not direct and required caution in the analysis of the results.

This latter GSMN, *i*AL1006, included 849 unique metabolites (*i.e.* after removing compartment information), of which 554 (62%) are annotated with an InChI. Conversion of these identifiers to InChIKeys [52] yielded 512 different first sections InChIKeys (*i.e.* the first section of InChIKey corresponds to the structure without stereochemistry). The *i*Prub22 reconstruction is associated with InChIKeys identifiers for 3,651 (67%) of its metabolites, with 3,382 being unique in their first section. A comparison of the InChIKeys revealed that 24 metabolites were exclusively present in *i*AL1006, while 488 metabolites were shared between *i*Prub22 and *i*AL1006. Among these 488 metabolites and continuing to disregard stereoisomerism, their first InChIKey section allows retrieving 617 compounds in *i*Prub22. Interestingly, 6 of these compounds were classified as "a secondary metabolite" according to the MetaCyc ontology.

Additionally, Prubens contained 2,594 unique metabolites, of which 2,415 (93%) possess a MetaCyc identifier (v23.0). Among these compounds, 2,377 were found in *i*Prub22, and only

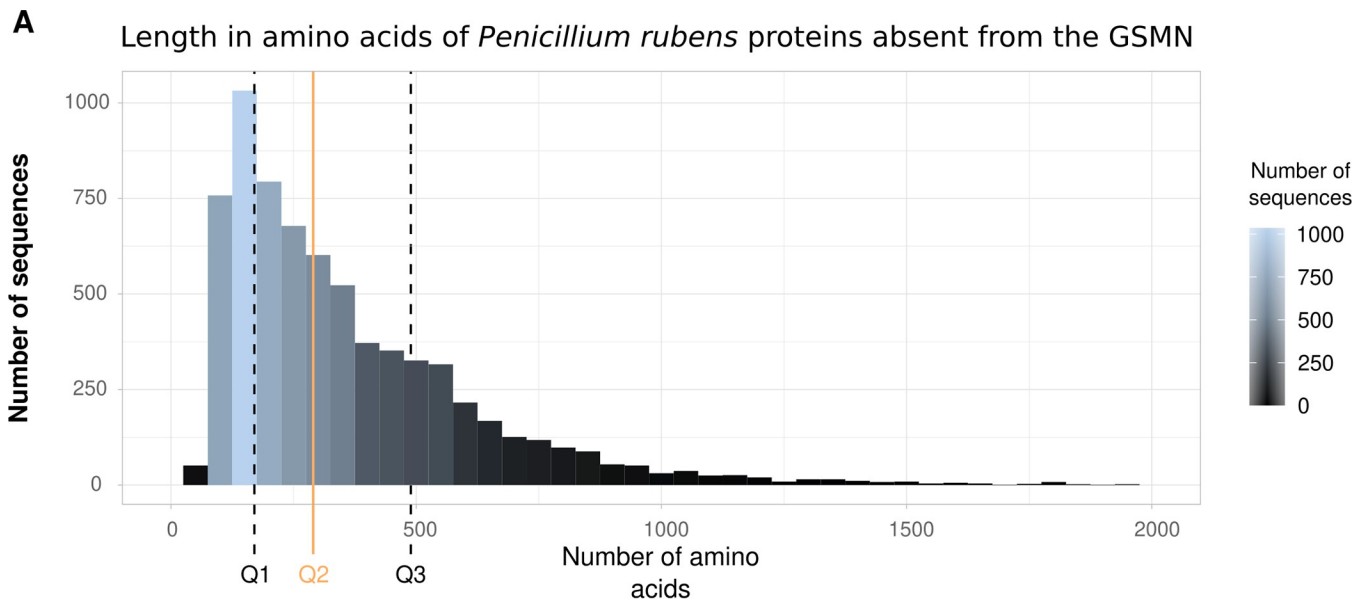

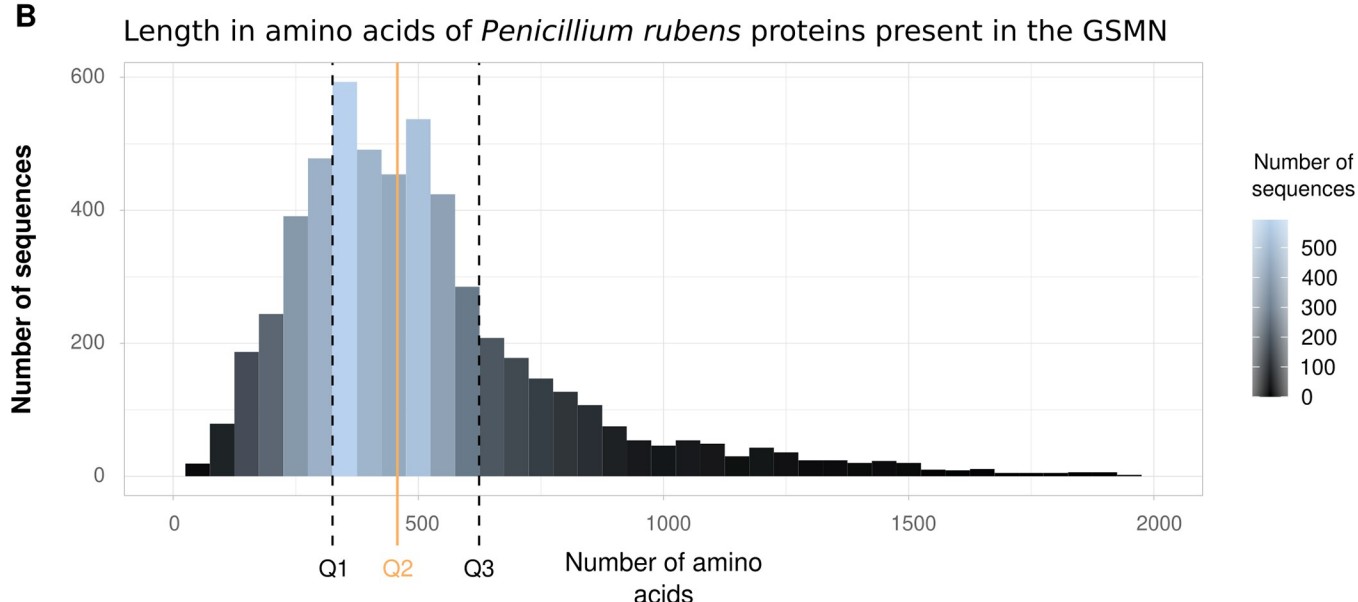

**Fig 5.** Amino acid length of *Penicillium rubens* sequences not included (A) and present (B) in the reconstruction. Median (continuous orange line ■) and quartiles (black dot lines) are shown. A range of 50 amino acids is displayed. (A) The average size of the 6,884 *P. rubens* sequences is 371 amino acids, and the median is 286. Measuring intervals: [15(min)-169];] 169–286];] 286–480];] 489–3,852 (max)]. (B) The average size of the 5,703 *P. rubens* sequences is 539 amino acids, and the median is 462. Measuring intervals: [28(min)-328];] 328–462];] 462–624];] 624–7,287 (max)].

38 were absent. Thus, *i*Prub22 encompassed 93% of Prubens' metabolites and was enriched with 2,825 additional entities. In the MetaCyc compound ontology, Prubens had 218 (8.4%) metabolites classified as "a secondary metabolite", 211 of which are shared with *i*Prub22, which itself contains 434 (8.4%) entities of this class.

Subsequently, our focus shifted towards understanding the predictive production capabilities of *i*Prub22 for a well-characterised list of specialised metabolites displayed in Fig 6. Thus,

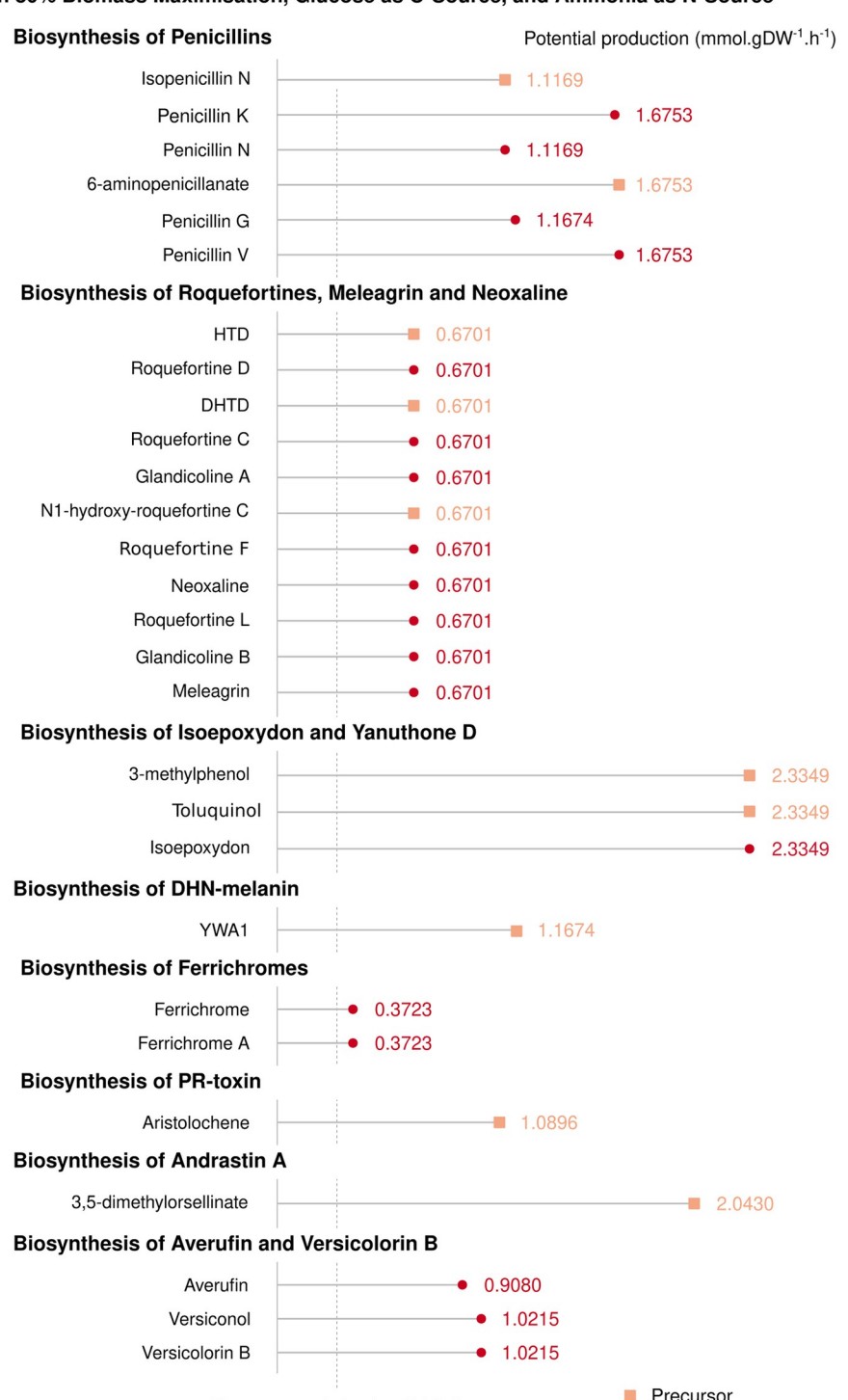

**Fig 6. Potential production of specialised metabolites in the *i*Prub22 model under reference conditions.** The figure displays the maximum flux of reactions associated with specialised metabolites biosynthesis (represented by red ●) or their closest known precursors (indicated as salmon squares ■) present in the *i*Prub22 model when the biomass is maximised to 80%. In addition to the imports required for biomass production (*i.e.* sulphur, thiamine, riboflavin, phosphate, oxygen, iron, glucose as C-source and ammonia as N-source), the uptakes of decanoate, phenoxyacetate and a general flavoprotein (NADPH-hemoprotein reductase) are open to allow the production of the depicted compounds shown.

we employed FVA to assess the potential production of compounds, whereby the maximisation of biomass was set at an arbitrary threshold of 80%. It is necessary to relax the constraints imposed on the system, allowing for the evaluation of flux variations in the exchange reactions of the targeted metabolites. As expected, in the absence of constraint relaxation, the flux variations were found to be null, indicating a lack of superficial production. This last point ensures the validation of the model's constraint definitions while simultaneously enabling an assessment of the potential production of specialised metabolites within the examined system. We evaluated eight biosynthetic pathways: seven were identified using the SMASH tool, and the eighth used mining reconstruction (*i.e.* pathways involving averufin and versicolorin B). In the context of databases such as MetaCyc, a pathway refers to a curated collection of biochemical reactions and associated genes or enzymes that are known to be involved in a specific biological process or metabolic. To provide a comprehensive analysis, we adopted a unified categorisation approach by grouping all intermediates, specialised metabolites, and by-products under the collective term "specialised metabolites" without differentiation between them. Consequently, our classification encompasses two distinct categories: specialised metabolites and precursors. Indeed, the production of metabolites, such as andrastin A, yanuthone D, and PR-toxin, poses challenges due to the absence of a MetaCyc identifier or associated reactions for their synthesis within the database. Therefore, it becomes necessary to focus on studying their precursors to gain insights into their theoretical distribution fluxes. Furthermore, the potential production of isoepoxydon, roquefortins and assimilates, averufin and assimilates requires the addition of Red-NADPH-hemoprotein reductase, an unbalanced compound with a radical in its empirical formula. This particular compound serves as a flavoprotein catalyst that facilitates the reduction of cytochrome. Additionally, the availability of phenoxyacetate is essential to unblock the flux of penicillin V, while the import of decanoate or octanoate was essential for the production of penicillin K.

**Interoperability.** To ensure the interoperability of *i*Prub22, the identifiers of metabolites and reactions were kept consistent with MetaCyc and cross-referred through MetaNetX [53]. We focus on various annotation enhancements (*e.g.* identifiers interoperability of metabolites, reactions, and genes between known databases) and on the addition of Systems Biology Ontology terms [54] (SBO) to match the standards established *via* MEMOTE. SBO annotations provide semantic information that characterises the model components [55,56]. A relatively low match rate on the BiGG (13%) and KEGG (43%) historical reconstruction databases was noted concerning metabolites annotations. On the other hand, more than two-thirds of the metabolites had a chemical identification (*e.g.* ChEBI, PubChem). Details are given in Table 2, and *i*Prub22's different characteristics are available in Supporting Information in the form of a MATLAB live script (S7 File). The reconstruction presented here is rated with a MEMOTE score of 74% [54]. Finally, based on these various annotations, it remains possible to complete them with mass spectral information from the MassBank [57] and GNPS [58] mass spectral databases for further identification in real biological samples. Based on a mapping from the InChIKey, 508 metabolites were associated with at least one spectral data.

Finally, interoperability between tools was ensured by upgrading the network distribution format to comply with SBML Level 3 [59].

## Discussion

### The strengths and limitations of reconstruction automation

Since the publication of a reference protocol for the reconstruction of GSMNs [60] and the emergence of omics data, numerous microbial models have emerged, essentially of bacteria. Methods established primarily from microorganisms have been automated [21,61], allowing

**Table 2. Details of the GSMN enrichment (nature and number of metadata).**

| Metabolites | | |
|---|---|---|
| Entities number | | 5,464 |
| ■ **Structure identifiers** | | |
| ○ InChI | 3,886 | 71% |
| ○ InChIKey | 3,887 | 71% |
| ○ SMILES | 5,276 | 97% |
| ○ Molecular weight | 5,270 | 96% |
| ■ **Database identifiers** | | |
| ○ BiGG | 717 | 13% |
| ○ BioCyc | 5,441 | 99.6% |
| ○ CAS | 959 | 18% |
| ○ ChEBI | 3,568 | 65% |
| ○ Chemspider | 1,487 | 27% |
| ○ DrugBank | 237 | 4,3% |
| ○ HMDB | 1,549 | 28% |
| ○ KEGG | 2,371 | 43% |
| ○ KNApSAcK | 96 | 1.8% |
| ○ LIPIDSMAPS | 138 | 2.5% |
| ○ MetaboLights | 1,052 | 19% |
| ○ MetaNetX | 5,436 | 99.5% |
| ○ ModelSEED | 4 | 0.07% |
| ○ PubChem | 3,693 | 68% |
| ○ SABIORK | 12 | 0.22% |
| ○ SwissLipids | 26 | 0.48% |
| ○ UMBBD | 50 | 0.92% |
| ■ **SBOTerm** | | |
| ○ Simple chemical SBO:0000247 | 5,464 | 100% |
| **Reactions** | | |
| Entities number | | 5,919 |
| ■ **Database identifiers** | | |
| ○ BiGG | 4 | 0.07% |
| ○ BioCyc | 5,162 | 87% |
| ○ Brenda | 4,343 | 73% |
| ○ KEGG | 2,470 | 42% |
| ○ MetaNetX | 5,436 | 92% |
| ○ Rhea | 2,873 | 49% |
| ○ Seed | 1,329 | 22% |
| **SBOTerm** | | |
| ○ Biochemical or transport reaction SBO:0000167 | 3 | 0.05% |
| ○ Biochemical reaction SBO:0000176 | 5,280 | 89.20% |
| ○ Translocation reaction SBO:0000185 | 60 | 1.01% |
| ○ Exchange reaction SBO:0000627 | 228 | 3.85% |
| ○ Demand reaction SBO:0000628 | 37 | 0.63% |
| ○ Biomass production SBO:0000629 | 1 | 0.02% |
| ○ ATP maintenance SBO:0000630 | 1 | 0.02% |
| ○ Sink reaction SBO:0000632 | 1 | 0.02% |
| ○ Transport reaction SBO:0000655 | 53 | 0.90% |
| ○ Active transport SBO:0000657 | 38 | 0.64% |

(*Continued*)

**Table 2.** (Continued)

| Metabolites | | |
|---|---|---|
| ○ Passive transport SBO:0000658 | 108 | 1.82% |
| ○ Symporter transport SBO:0000659 | 21 | 0.35% |
| ○ Antiporter transport SBO:0000660 | 2 | 0.03% |
| ○ Spontaneous reaction SBO:0000672 | 86 | 1.45% |

simultaneous and multiple eukaryotic GSMN creations [34,39]. In the meantime, several tools [61,62] have also been developed, raising interoperability issues and the well-known need for standardisation. Thus, AuReMe [63] was chosen for its data traceability and network visualisation capabilities. This toolbox merges data from functional annotation, orthology search from previously published models and manual expertise. Traceability of the reaction origin has a double objective. It provides information both on evolutionary and phylogenetic concepts, thus highlighting the core genome, which mainly characterises the genes involved in essential intermediary metabolism, DNA replication and repair, and transcription and protein synthesis. Additionally, traceability provides a quality estimation of the reconstruction by giving a 'confidence weight' to the reactions. This manual curation can be performed in two ways, focusing either on reactions or on genes. Thus, all of the information presented in Fig 1 and S7 Fig in the S1 File are clues to guide the refinement of the draft toward obtaining a GSMN. In addition, the complementarity between functional annotation, orthology search and external sources addition during the reconstruction process is also visible in S8 Fig in the S1 File. Hence, the used approaches and data complementarity are the heart of the GSMN reconstruction process for *P. rubens*. Therefore, is there a legitimate interest in cross-referencing all of this information which is fundamentally and originally the identical mechanism result, which is sequence homology?

The first question that can be asked concerns the choice between carrying out a reconciliation of pre-existing data or a *de novo* reconstruction. As described by Oberhardt *et al.*, the reconciliation process implies that for the two previously reconstructed networks, if there is "no discriminating biological evidence for a given reaction to being included in one reconstruction but not the other, the reactions should be identically included in both". The struggle is real since working with reconstructions using incompatible or "homemade" identifiers is a significant source of error and a complicated task (S6 Fig in S1 File). However, in terms of network evolution perspectives, the consensus and systematic comparison of these data could lead to improvements (*i.e.* refining genomic information by updating the GPR associations) for future manual curation.

These comparisons provide information on the complementarity of the approaches and the sensitivity/specificity of the GPR associations. Thus, one of the limits of the automated reconstruction lies in the assignment of genes for non-specific reactions (*i.e.* a large number of genes being associated with such type of reaction). Moreover, intra-operability and especially identifiers durability as the databases are updated is sometimes not guaranteed. For example, of the 1,291 reactions from PchCyc, 79 (6%) have an incompatible identifier with MetaCyc version 23.0. For these reasons, few tools currently exist that allow automatic comparison of these networks, although efforts are being made in this direction [64]. As a result, relatively few model organism reconstructions benefit from pre-existing data [19,20], and when they do exist, they require a significant community effort [18].

Another example of information loss is related to database mapping, which arises in particular when integrating subnetworks obtained by orthology search. During the creation of the

orthology subnetwork, more than half of the reactions were discarded due to a lack of Meta-Cyc-compatible identifiers. These reactions, supported by genomic data, could, therefore, constitute a preferred pool for future completion of the GSMN (*i.e.* addition of reactions or refinement of existing GPRs). For example, from the 546 lost reactions during the mapping on *N. crassa*, 505 were allocated to metabolite transport and had model-specific identifiers. Only manual curation will allow for the recovery of such information. Thus, this known interoperability problem (*i.e.* lack of MetaNetX [53] identifiers or model-specific identifiers) highlights the need for uniform writing standards both at the database level and in the intrinsic writing of models. Also, this points out the issue of the relevance of reconciling the subsequently reconstructed networks. While these networks carry relevant genomic information, their integration is tricky, as shown in the section on modifications of the reconstruction to obtain a viable flux model.

The second question is whether orthology search should be worth using in addition to functional annotation. Are these two approaches redundant or complementary? As orthologous genes arise by speciation events, these groups of genes are assumed to share identical biological functions [65]. By definition, the search for orthologous sequences allows the metabolism section detection common to organisms (*i.e.* consensus), thus highlighting their evolutionary history. Constitutive metabolism will therefore be mainly targeted by this type of approach since it is shared between various species. On the other hand, due to its specificity, specialised metabolism can only be incorporated into the reconstructions by functional annotation or, if it fails, by manual curation. The functional annotation approach should, therefore, cover the one performed by orthology search. However, 16% of the reactions in the draft still come exclusively from one or more subnetworks resulting from the orthology search. Therefore, this method also helps to benefit from the manual expertise within the reconstructions used for orthology search, as it would appear that the information they contain is not systematically referenced in the functional annotation database (*i.e.* 764 reactions are exclusively specific to the orthology subnetwork–Fig 1). In addition, one would also expect the most informative templates to be the most phylogenetically related. In the case of the *i*Prub22 reconstruction, most information obtained by orthology search came from *S. japonica*. Therefore, the most informative template for the reconstruction is not fundamentally the most biologically relevant. Since the selected GSMNs were published between 2010 and 2018, and *S. japonica* GSMN was the most recently reconstructed network using a protocol very similar to the one used in this study, this reflects a lack of details in the past and present data rather than proper evolutionary concepts.

As mentioned above, the specialised metabolism of an organism can probably only be raised by functional annotation. What about its presence within the proposed reconstruction? This point can be illustrated by focusing on two compound families: roquefortines and penicillins. First, roquefortines and meleagrin are indole alkaloids produced by several *Penicillium* and *Aspergillus* species from tryptophan and histidine. Knowing that all the information related to the synthesis of these compounds is encoded in MetaCyc (since version 20.5) and is grouped under the PWY-7609 super-pathway, why is there no trace of them, even partial, in the draft? As described in Guzmán-Chávez *et al.* [10], roquefortines and meleagrin are catalysed by 13 reactions mediated by 6 enzymes/genes. Since their pathways are not described in previous reconstructions or orthology templates, the only possibility to incorporate this specific pathway in our reconstruction is from the functional annotation subnetwork. However, although the 6 genes involved in the biosynthesis of these compounds were present in the draft and encoded between 2 and 20 reactions, no reactions related to the pathways concerned were found in the draft. Backtracking its absence, through traceability provided by AuReMe, indicated that this absence is related to an error in the PathoLogic module use. It unintentionally

and automatically pruned 9 genomic sequences whose 6 belong to the roquefortines pathways. Concerning the second example, within MetaCyc, the reactions leading to different penicillins biosynthesis are listed in 3 pathways: PWY-5629 (isopenicillin-N biosynthesis– 2/2 reactions found in the draft); PWY-7716 (penicillin G and penicillin V biosynthesis– 3/4 reactions found in the draft); PWY-5630 (penicillin K biosynthesis– 1/1 reactions found in the draft). Although their names vary from one database to another and within the literature, the genes responsible for the production of these compounds were identified as pcbAB or ACV or ACVS (*Pc21g21390*); pcbC or IPN or IPNS (*Pc21g21380*); pcbDE or iAT or AAT or PenDE (*Pc21g21370*); phl or pclA (*Pc21g14900*). PenDE, ACVS and phl were discarded by PathoLogic during reconstruction like the roquefortines biosynthetic pathways genes. It would appear that discarded genes and associated reactions are deleted if, and only if, they are not supported by any other gene. For example, the following two reactions are catalysed by PenDE: RXN-1702 and RXN-17100 (*i.e.* production of penicillin G and interconversion of penicillin G to penicillin V, respectively). For the first reaction, PathoLogic infers two sequences (*i.e.* one corresponding to PenDE based on the EC-number and another based on the GO annotation). Since this reaction is supported by two genes, the tool does not perform any pruning step. Conversely, the second reaction is only supported by PenDE and is consequently excluded from the reconstruction, as are all reactions related to the synthesis of roquefortines. Nevertheless, the reconstruction methods' complementarity allows us to recover almost all of the penicillin biosynthetic reactions, even if the GPRs are not precise enough. Moreover, the genes associated with penicillins synthesis, *Pc21g21380* (IPNS), *Pc21g21370* (PenDE), and *Pc21g14900* (phl), were present in the draft and supported between 2 and 12 reactions. However, *Pc21g21390* (ACVS) was absent from the draft even though the reaction it catalyses was found in it and supported by 10 other genes found by orthology. These results raise questions about the relevance and accuracy of the automatically deduced GPRs associations and, thus, more broadly, about the quality of the networks used for reconstruction through orthology. Therefore, in addition to the entire PWY-7609 pathway for roquefortine biosynthesis (S3 Table in S1 File), manual curation was limited to the inclusion of the RXN-17100 reaction (PenDE-mediated interconversion of penicillin G to penicillin V) and the ACVS gene for the 6.3.2.26-RXN reaction (the first reaction inducing isopenicillin-N synthesis). Even if GPRs' specificity and accuracy are essential for model simulations (*e.g. in silico* knock-out), verbose GPRs within a reconstruction remain sources of information to explain evolving concepts. Thus, GPRs associations cleaning S4 Table in the S1 File) was not performed.

The third and last point concerns the limits of reconstruction automation, notably the gap-filling steps. Gap-filling makes it possible to complete the pathways from a structural point of view. As this process is based on heuristics and the absence of genomic data, additional results from such tools must be done sparingly and carefully. Here, sets of targets were used to perform gap-filling against proper subsets of MetaCyc depending on the purpose (*e.g.* preferential selection of spontaneous or fungi reactions). Iterative processes were performed with the repair networks and targets from most to least relevant to achieve 'reasoned gap-filling'. Nonetheless, beware that performing a complete gap-filling, *i.e.* using the set of putative metabolites present in our reconstruction against the entire MetaCyc database, is absurd from a biological and evolutionary point of view. This set was used only against controlled versions of repair networks (*i.e.* for the detection of spontaneous reactions and for the last round of gap-filling whose supporting genomic sequences are unknown in MetaCyc). By reasoning in this way, it is possible to give an additional confidence weight to almost half of the gap-filling reactions (S12 Fig in S1 File). As shown in the results presented in Table 1, gap-filling improves the connectivity of the GSMN. The semi-automated process for the draft generation allows topologically reach 86% and 22% of the selected targets belonging to the constitutive and specialised

metabolisms, respectively. These numbers demonstrate the methods' efficiency for the automatic reconstruction of primary metabolism. It also points out how far we still have to go with regard to specialised metabolism. Furthermore, it should be remembered that, as described in Dusad *et al.* [66], topological approaches are limited by two well-known issues, namely hub metabolites and reactions reversibility. This last point will be discussed later in the section on the reconstruction evolution to the flux model.

Finally, what parameters should be considered as a plausible possible model, as close as possible to biological reality (*e.g.* addition, modification, deletion of reactions and constraints)? This point should be addressed at various scales, from the most generic (*e.g.* addition of layers of information related to compartmentation, spontaneous reactions and exchange reactions) to the most specific and precise (*i.e.* refinement of metabolic pathways with a focus on the nature of GPRs associations). Due to their nature, these steps often require manual expertise and thus highlight the limits of such automation. As *i*Prub22 is intended to be improved in this sense with feedback from the scientific community, a user-friendly wiki is available to facilitate communication and improvement for future versions. The wiki associated with the GSMN offers a less cumbersome virtual alternative to the efforts developed during the nonetheless indispensable processes of community meetings over the past years (*e.g.* jamboree or workshop) [67–69].

## GSMN quality

The GSMN origins date back to 1999, with the first model publication for the bacterium *Haemophilus influenzae* [70]. Thiele and Palsson published a protocol for reconstructing metabolic networks in four key steps ten years later, which is still used as a reference today [60]. However, in the history of GSMN reconstruction, the development of uniform criteria for judging the quality of a model is relatively recent, encouraging the tendency to prefer quantity over quality [71]. Except for the MEMOTE test suite [54], there are still few tools or measures available to assess the completeness of a GSMN. Thus, the GSMN presented here has been reconstructed following, among others, these three minima and general criteria defined by the work of Monk *et al.* [71]:

- the extent of manual curation: what aspects and means are used to refine the model? This point was addressed in the previous section.

- metabolic coverage: what is the metabolic space, *i.e.* the genome proportion, covered by the data present in the GSMN?

- the use of established standard operating procedures: what are the means and standards of writing to be used to encode the information? This step is independent of biological knowledge and allows the GSMNs application scope to be extended.

With the improvement of methods and the data proliferation, the *P. rubens* GSMNs evolution underlines a trend towards improved metabolic coverage over the years that seems to be enhanced substantially. The function of an enzyme is intrinsically dependent on its three-dimensional shape, *i.e.* structure and folding, which in turn is induced by the amino acid sequence of which it is composed. It should also be noted that eukaryotic proteins are generally longer (*i.e.* median size of 360 amino acids [72]) than those of other living organisms. Therefore, the fewer amino acids a *P. rubens* sequence contains, the less likely it is to carry a functional catalytic domain. The sequence size distributions presented in Fig 6 shows that of the sequences discarded from the GSMN reconstruction, almost half are less than 300 amino acids in length. The remaining half is of sufficient size to encode functional enzymes and constitute

a reservoir/pool of candidates for future improvement of the network. This set of genes with unknown functions and probably specific to the studied organism offers research perspectives for an improvement of future models (*e.g.* unlocking the topological accessibility of certain dead-ends).

Computer modelling is one of the means used by Systems Biology to understand, organise and integrate large quantities of data from diverse sources (*e.g.* molecular, cellular, *etc.*). Relying on ontologies (*i.e.* set of controlled vocabulary) allows the exchange, interoperability and reuse of the models produced. This step, independent of model functionality, can occur at all stages of its life cycle. One aspect related to computational interoperability is associated with the annotation of the model entities in terms derived from the 'Systems Biology Ontology' (SBO) [55,56]. This type of annotation is not specific to metabolism modelling. However, it allows, like any ontology, to generate a classification and a hierarchy of the data making them easier to interrogate for the user since controlled vocabulary use is part of good practice recommendations for computer modelling. In addition to the distribution format, network sustainability depends on its usability. It is well known that interoperability between databases is a delicate task. Therefore, the most exhaustive annotation of the entities in the model is essential and is now an undeniable criterion for increasing the quality of a network [23,25]. Firstly, when these annotations belong to specific databases such as KNApSAcK (*i.e.* a database dedicated to specialised metabolites), they constitute an entry point for model exploration. For information purposes, *i*Prub22 contains 80 metabolites annotated with an identifier from this database. Secondly, the molecule's structural identifiers allow the entities to be identified and used as an anchor for querying other databases. For instance, using an InChIKey mapping considering only the first identifier section, thus disregarding the stereochemistry and compound's charge, 9.8% of GSMN metabolites are linked to mass spectral information retrieved from the GNPS and MassBank databases. This coverage is lower than the average 40% detected by Frainay *et al.* work [73]. This confirms the ever-limited aspect of obtaining and integrating metabolomics data acquired in given organisms within GSMNs. Such difficulty could be one more problem in validating GSMN behaviour, at the chemical level, with respect to biological observation.

However, as mentioned above, MEMOTE [54] is a suite of tests aimed at qualifying a reconstruction or model by providing a summary of its characteristics both in terms of topology and flux. Its quality score considers the annotation content of the model. For example, a complete annotation in SBO terms guarantees a quality of 25%. This tool provides an easily accessible resource aiming toward model standardisation for detecting blocking points and/or abnormalities, thus helping manual curation. The GSMN we propose possesses a MEMOTE score of 74% against 31% and 16% for the 2013 and 2018 networks, respectively. Nevertheless, due to the biases taken during the reconstruction, such as the creation of artificial genes to help their identification, the systematic conservation of all data supported by genomic information and the annotation of some SBO terms not currently monitored, the 74% score is slightly underestimated (the MEMOTE report is available in S8 File). Furthermore, this tool is designed and optimised to evaluate models that preferentially respond to the SBML3 FBC2 format [59,74]. Since the previous networks were not encoded with the flux balance constraints (FBC) package, some characteristics cannot be evaluated properly, and thus, their MEMOTE results should be put into perspective. This is particularly true for analyses related to genes and their products, as this notation was only introduced in Version 2 of the FBC package.

The recurrent non-reproducibility of results, known as the reproducibility crisis [75], affects all areas of science and *in silico* modelling is no exception [76]. In fact, limited interoperability between software may yield incorrect models parsing, which can lead to errors or misunderstandings when analysing GSMNs [77]. Thus, the proposed model was standardised in

the most recent Versions of SBML and its extension package flux balance constraints (FBC). However, even if the emphasis on reconstruction has been placed on the maximum transparency and traceability of data acquisition, with the constant evolution of databases and tools, providing all the information for the strict whole model reproducibility remains challenging. For this reason, GSMNs are regarded as knowledge state snapshots about an organism at the precise moment of its reconstruction, so a particular effort to keep on watch would be necessary to ensure their sustainability. Ultimately, the main objective of this reconstruction is to propose a resource grouping of known data for *P. rubens* based on these three key complementary notions: traceability, accessibility and visibility, focussing additionally on the exploration of specialised metabolisms' presence.

## From reconstruction to a model that simulates environmental exchanges

Transforming a genome-scale reconstruction into a computational model for predicting properties and behaviour is a considerably time-consuming process that requires expertise. Even if steady-state constraint analyses are neither informative on the concentration of the metabolites nor on the temporal dynamic system aspects, it is still the more appropriate tool for exploring complex biological systems. These analyses are based on linear optimisation theories and constraints implementation based on three aspects: physicochemical (*i.e.* steady-state, stoichiometry, thermodynamics), topological (*i.e.* spatial modelling of intracellular and extracellular compartments) and environmental (*i.e.* various nutrient media simulations defined by exchange and transport reactions) [78]. The previously obtained reconstruction is used as a backbone for mathematical approach modelling, which will perform *in silico* predictive hypotheses-driven. However, the transition from the reconstruction model development to the fluxes models requires various adaptations, both computational and biological.

The uncertainty inherent in the GSMNs predictions is influenced by the numerous tools, methods and algorithms that exist for both the reconstruction phase (*i.e.* network structure) and the analysis phase (*i.e.* simulation results). However, the specific sources and magnitudes of these uncertainties remain complex to quantify, which presents challenges for evaluating the relevance and reliability of model predictions. In this respect, the recent great review by Bernstein *et al.* [24] identifies five uncertainty areas surrounding the reconstruction and analysis of GSMNs. These fields include (1) genome annotation, (2) specification of environmental conditions, (3) formulation of biomass equations, (4) network gap filling, and (5) flux simulation. Their work highlights the need to assess, communicate and understand the uncertainty sources associated with a model to discuss their impact on the relevance of predictions resulting from the model analysis. As functional annotation and gap-filling have been dealt with previously, we will focus here on the specification of the environment and the formulation of the objective biomass function. We would also like to stress that the focus of the work presented in this paper is on reconstruction considering specialised metabolism, and thus, we propose preliminary work on the network analysis of the flux distribution. Finally, an insight into the potential avenues for improving the model will be discussed in this section.

The biomass reaction is a commonly used objective function in constraint-based modelling, including FBA and FVA, that simulates the growth of a specific organism. It identifies essential growth compounds and assigns weights based on their occurrence, normalised to represent the dry-weight biomass of 1 gram. Understanding and optimising the growth of organisms requires defining the biomass reaction and its subsystems, which correspond to the synthesis of macromolecules from building blocks and precursors. Experimental values for growth-associated ATP maintenance (*i.e.* GAM) and non-growth-associated ATP maintenance (*i.e.* NGAM) are added to the reaction, representing the energy required for macromolecule

polymerisation during growth and for maintaining the organism's viability, respectively [60]. However, in the absence of specific data, the biomass composition of a model organism is often used as a template, generating biases in the interpretation and aggregation of results when comparing GSMNs [24]. *Penicillium rubens* is a well-studied model organism, and experimental data indicate that its relative macromolecule content is distributed as follows: 45% proteins, 25% carbohydrates (including 22% cell wall compounds and 3% glycogen), 5% lipids (including 0.5% fatty acids, 1% sterol esters, and 3.5% phospholipids), 9% nucleic acids, and 8% each of ASH and the soluble pool [79,80]. As the biomass reaction in *i*AL1006 was constructed based on these experimental data [26], the same reaction is used in *i*Prub22. However, *i*Prub22 lacks several subsystems found in *i*AL1006 that regulate the biosynthesis of fatty acids (*i.e.* r1434), phospholipids (*i.e.* r1460), lipids (*i.e.* r1464), sterol esters (*i.e.* r1462), and glycerides (*i.e.* r1461). These subsystems could not be included in *i*Prub22 due to the absence of compatible or relevant MetaCyc identifiers. Finally, refining or improving the biomass reaction for more accurate growth prediction can be achieved through sensitivity analysis, identifying critical reactions and key parameters for maximum biomass yield, or experimental measurements to determine the exact quantities of components needed for growth, allowing adjustment of the biomass reaction coefficients. For instance, the BOFdat workflow [81] may offer a prospective solution to incorporate lipid monitoring, which is currently not present in *i*Prub22.

Defining the chemical composition of the environment for metabolic modelling is a delicate task fraught with uncertainties, as described by Bernstein *et al.* [24] (*e.g.* inconsistent media definition in specific databases or undefined chemical input in experimental settings). However, environmental specification is a critical component of flux analyses and serves as the cornerstone for such investigations. As filamentous fungi are known to possess cellular machinery that enables extracellular nutrient absorption, uptake reaction selection was carried out by focusing on extracellular metabolites. Specifically, we integrated reactions from the well-curated *i*AL1006 model and complemented them with information on the extracellular localisation of gene products (*e.g.* inclusion of reversible or non-reversible transport reactions between the intracellular and extracellular compartments, regardless of whether they are artefactual). Thus, we adopted the "seed" approach (*i.e.* metabolites occurring naturally in the environment of the organism studied and essential input for the GSMN) to generate biologically relevant models that have the potential to enhance our understanding of metabolic processes and facilitate future investigations. Indeed with 185 uptake reactions included in the reconstruction, *i*Prub22 offers a broad range of possibilities for modelling simulations. Moreover, proposing diverse combinations will give clues on organism behaviour by identifying aspects of *i*Prub22 predictions that exhibit either high sensitivity or resilience to changes in the environmental composition. Finally, once the nutrient nature has been defined, the subsequent step entails relying on experimental data to establish proper uptake bounds for flux simulation analyses.

Then, the modifications performed to deal with problems encountered when converting the reconstruction to a flux model focus mainly on updating and correcting (1) the directionality of reversible reactions (*i.e.* adding constraints by closing reactions whose directionality remains unknown on MetaCyc), (2) the redundancy contained in the reconstruction and (3) the identification of blocked reaction.

The redundancy point is addressed at different levels. On the one hand, automatic and manual curation of duplicated reactions is carried out (*i.e.* reactions with the same reactants and products). The presence of such reactions is explained by the duplicate's existence within the databases and by the data reconciliation contribution (*i.e.* addition of reactions whose unbalanced equations do not allow their automatic detection). On the other hand, there is redundancy within the networks linked to varying metabolite names. Indeed, the same chemical entity may have several forms, and thus identifiers, that coexist within the model without

having a bridge reaction between the different forms. Examples include the case of coexisting linear or cyclised states of molecules (*e.g.* carbohydrates), the chirality encoding as a function of the different stereoisomers, or the use of more generic concepts with compounds class against single chemical species (*e.g.* molecules ontology). For instance, if D-galactopyranose is a compound class composed of ALPHA-D-GALACTOSE and GALACTOSE (*i.e.* beta form) and if an interconversion reaction of alpha and beta form exists, it seems reasonable to replace D-galactopyranose by ALPHA-D-GALACTOSE. From a computational perspective, this identifiers multiplication, or impoverishment when the annotation is too specific, leads to ultra-connected or under-connected network sections. Uniformity of identifiers on parent or child terms can thus reduce such artefacts. Concerning Blocked reactions in a metabolic model, their presence indicates infeasibility and highlights potential errors, connectivity issues, lack of necessary metabolite inputs, or gaps in our understanding of the metabolism. These reactions are unable to sustain any metabolic flux under steady-state conditions and require investigation and correction to ensure model accuracy. As these corrections directly influence the outputs of these constraints-based analyses, all the modifications applied to the model generation are reported as additional data. By employing these constraint-based modelling techniques, we gained valuable insights into the metabolic behaviour and potential of *i*Prub22, further validating its utility as a predictive tool for *P. rubens* metabolic engineering and biotechnological applications.

The above-mentioned modifications (S7 File) have yielded a model that exhibits diverse growth behaviours in response to variations in 35 different carbon sources and 35 different nitrogen sources (Fig 3 displays a sample of the results, and the complete data are provided in the S6 File). This approach provided valuable insights into the models' performance and their responses to varying nutrient availability scenarios. Interestingly, the *i*Prub22 model demonstrates its ability to reflect some observed behaviours with *P. rubens*. In accordance with Nielsen *et al.* observations [80], it is established that *P. rubens* requires a source of carbon, such as sugars, polysaccharides, organic acids, lipids, or certain amino acids, and a source of nitrogen, which can be provided by organic or inorganic compounds like ammonia, nitrate, or nitrite, to support its growth. However, it is worth noting that the carbon availability for respiration is relatively more limited compared to nitrogen. Thus, we have arbitrarily set the nitrogen limit at one-third of the carbon limit. Furthermore, it is observed that this fungus cannot grow in the nitrogen source absence. Besides, the amino acid L-cysteine alone is not a nitrogen source sufficient for the growth of *P. rubens*. These various behaviours are expressed by *i*Prub22. When ammonia and glucose are present, the model predicts a growth rate of 0.2944 mmol. gDW-1.h-1, which falls slightly outside the range of 0.14 and 0.22 h$^{-1}$ established by the work of Grijseels *et al.* [50].

Nevertheless, by recalibrating the uptake boundaries, it will be possible to better align with the experimental data. However, it is crucial to acknowledge the presence of more divergences between the predictions of the model and the experimental findings. These disparities serve as valuable insights for further refining and improving the model, ensuring its accuracy and reliability in predicting biological phenomena. The study conducted by Allam *et al.* demonstrated that among the tested nitrogen conditions, the highest growth rates were observed in the following decreasing order: hypoxanthine, adenine, sodium nitrate, xanthine, urea, and ammonia [82]. However, in our model, the optimal growth conditions were found to be with xanthine, hypoxanthine, adenine, ammonia, urea, and nitrate, as illustrated in Fig 3. Moreover, the *in silico* prediction of biomass flux on sucrose [49] surpasses the experimentally measured growth rate by a factor of 3.5. On the other hand, contrary to what the experimental data suggest [80], the model appeared not to be able to simulate growth on media whose carbon source is D-isocitrate, cellulose or acetate.

Finally, the model expresses mixed responses when amino acids can serve as both carbon and nitrogen sources. *In vivo*, the fungus shows a greater preference for amino acids that can rapidly degrade into a carbon source in one or two steps. These amino acids can be classified into three groups: L-glutamine, L-asparagine, L-arginine, and L-proline (group 1); L-glutamate, L-aspartate, L-alanine, and L-ornithine (group 2); L-histidine, L-glycine, L-isoleucine, L-lysine, L-leucine, L-methionine, L-phenylalanine, L-tyrosine, L-threonine, and L-valine (group 3). The model cannot offer a solution when L-histidine, L-tryptophan, L-methionine, L-isoleucine, L-leucine and valine are used as nitrogen and carbon source; amino acids from the third group, which are less suitable for the fungus. However, L-phenylalanine, L-tyrosine, and L-lysine, which also belong to this group, are the amino acids for which the simulated growth potential of *P. rubens* is the highest. These discrepancies suggest that the model may have insufficient constraints, allowing an exaggerated increase in the growth rate. Consequently, growth differences in different media may be attributed either to the high metabolic robustness of the fungus or to the extreme pathways' presence, such as futile cycles inside the model [83]. Thus, one of the most plausible hypotheses for correcting this inaccuracy is to investigate the presence of thermodynamically infeasible cycles (*e.g.* type II or III extreme pathways, also known respectively as futile cycle or internal cycle) within the model [84]. In metabolic network modelling, thermodynamically infeasible cycles are a group of reactions that violate the laws of thermodynamics when active (*i.e.* unbalanced production or consumption). As thermodynamically infeasible cycles do not generate advantageous metabolites while consuming energy, they are not considered feasible pathways within the metabolic network. The presence of these cycles is generally attributed to inaccuracies or errors in the reconstruction process, stemming from incomplete knowledge of metabolic pathways or oversimplified assumptions regarding, for instance, reaction reversibility or the absence of regulatory processes. The most commonly used method for internal cycle detection is the II-Cobra method [85], and other algorithms have emerged over the years, such as the CycleFreeFlux algorithm [86]. GlobalFit [87] identifies erroneous energy-generating cycles (*i.e.* cycles that charge energy metabolites without a source of energy since they consume cofactors to generate motive power), which are futile cycles that are more difficult to identify and highly prejudicial to the simulations. Therefore, the presence of infeasible cycles can significantly impact the predictive capability of models, making it essential to identify and eliminate them to enhance the accuracy and reliability of the models.

Finally, a significant improvement of the proposed model will lie in GPRs verification and correction. For reconstruction, the false positive presence within a GPR carries information (*i.e.* genomic sequence encoding an enzyme with a similar mode of action). Thus, the cross-referencing of false positives with GPR and gap-filled reactions would enable the pool creation of preferred candidates to be interrogated to detect, for example, new functionalities or to highlight evolutionary concepts. On the other hand, from the computational model side, erroneous GPRs greatly complicate the analysis of knock-out results *in silico*. Incorrect predictions, linked to imperfect knowledge of the models studied, are an opportunity for biological discovery. Indeed, the abnormal behaviours of the model allow us to pinpoint the most relevant aspects of the metabolism and are, therefore, sources of hypothesis for addressing these failures. It is a valuable help in guiding experiments that often lead to new organism functionality characterisation.

## What about specialised metabolites?

The reference strain *P. rubens* Wisconsin 54–1255 is the result of a long classical strain improvement programme [11]. Various rounds of random mutagenesis have led to the

selection of a strain capable of producing higher quantities of penicillins, notably at the expense of other specialised metabolites production expressed by *P. rubens* NRRL 1951 wild-type strain [88]. Proteome analysis [89] and the study of these mutations [11] revealed lower expression levels for some genes related to specialised metabolism and the formation of non-functional proteins. The proposed GSMN expresses the potential for metabolite production carried by the genome and thus provides a platform for knowledge that can be enhanced from such data. However, it should be noted while the presence of a compound in the model offers a valuable starting point for further investigation, it does not necessarily guarantee its production within the model.

According to the classification of metabolites established by the MetaCyc ontologies, 13% of *i*Prub22 compounds would be associated with specialised metabolism. However, it is not uncommon to find metabolisms building blocks relative to constitutive metabolisms such as geranyl diphosphate or myoinositol in the specialised metabolites lists. This fact illustrates that the specialised metabolites concept, as understood by biologists or chemists, diverges from what is implemented in the databases. Thus, based on the information provided in the literature and the results of genome mining, the percentage of specialised metabolites contained in *i*Prub22 should be further minimised. It raises the question of what constitutes a specialised metabolite for these communities and highlights the constant need for annotation standardisation.

Fungal specialised metabolites' particularity lies in their biosynthetic genes being co-located and forming sets called "Biosynthetic Genes clusters" (BGCs). Today such clusters are commonly identifiable *via* genome mining, but many of their products remain unknown. The review by Iacovelli *et al.* [90] reports 33 essential biosynthetic genes encoding 10 non-ribosomal peptides synthetases (NRPSs), 20 polyketide synthetases (PKSs), 2 hybrid NRPS-PKSs, and 1 dimethyl-allyl-tryptophan synthetase (DMATS). Non-ribosomal peptides identified include fungisporin, roquefortine C, D, F, M, and N, as well as associated products such as meleagrin, glandicolins A and B, histidyltryptophanyldi-ketopiperazine (HTD) and dehydro-histidyltryptophanyldiketopiperazine (DHTD). This list of NRPSs is also enriched by the following three siderophores, coprogen, ferrichrome and fusarinin family compounds (*i.e.* being classified by their function instead of their biosynthetic pathways, here NRPSs). Among the polyketides identified are compounds belonging to the sorbicillinoid family and chrysogin. All these metabolites are well-documented, studied and commonly accepted in the literature [7,10,11,13,91].

However, the identification of BGCs and their end products slightly differed depending on whether fungiSMASH or antiSMASH was used (S5 Table in the S1 File). Thus, three BGCs coding for the following five compounds were found exclusively with fungiSMASH: naphtho-pyrone, chrysoxanthones A, B and C and depudecin. Concerning the antiSMASH results, the BGC related to penicillin synthesis emerged more exhaustively since isopenicillin N, phenoxy-methylpenicillin (penicillin V) and the precursor $\delta$-(L-$\alpha$-aminoadipyl)-L-cysteine-D-valine (ACV) were identified. In addition, BGCs leading to the synthesis of PR-toxin, neurosporin A, ACT-Toxin II, melanin, NG-391, aspercryptins and chaetoglobosins are also raised only with antiSMASH results. It is also worth mentioning that the similarity percentages (*i.e.* the confidence that can be placed on the actual presence of the BGC in the strain under consideration) can also be different from one version to another, as illustrated by the sorbicillin-encoding cluster.

Of the 42 BGCs identified using the fungiSMASH tool suite, 29% have yielded at least one distinct compound. However, only three of these BGC products (*i.e.* involved in the biosynthesis of penicillins, roquefortins, and initial steps of patulin) have been integrated into GSMN. Collecting and processing the information on specialised metabolites biosynthesis raises

several questions. Firstly, are the metabolic pathways leading to the biosynthesis of those NPs comprehensively elucidated and documented in the literature? Furthermore, if so, are these pathways indexed in the relevant databases such as MetaCyc? For instance, CobraMod [92] is a pathway-centric curation tool designed to enable the modification and extension of GSMN. This tool extracts metabolic information from BiGG, KEGG, and BioCyc databases. However, depending on the existence of biosynthetic pathways for the selected metabolites and their degree of completeness, what measures are envisaged to enhance the modelling accuracy? This point leads to the two lists creation, one containing the metabolites for which an identifier exists in MetaCyc, and the other composed of orphan metabolites for which, among other things, the closest known antecedents (*i.e.* precursors) will have to be determined. Subsequently, if the metabolic pathways are fully present, curation consists of verifying the actual presence or absence of the reactions in the draft and, if the information in the literature provides it, checking the corresponding GPR associations. The accuracy of the GPR associations is pivotal for the biological and computational interpretation of the *in silico* knock-outs. Thus, to conform with the literature data, manual curation was applied to *i*Prub22 for penicillin, meleagrin, roquefortine and the last known precursor of patulin produced [7]. On the other hand, if the metabolic pathways are semi-complete, like for yanuthone D [93] or absent, like for chrysogine [94], the analyses will focus on the closest known precursors, and further work will aim to find the most accurate method for integrating and generating the missing information. These gaps could potentially be resolved by adding modules or a single reaction, balanced and supported by all the genes involved in the biosynthesis of the compounds of interest. Recently an automatic pipeline called BiGMeC (Biosynthetic Gene cluster Metabolic pathway Construction) has been developed to automatically reconstruct metabolic pathways associated with BGCs, specifically targeting PKS and NRPS [95]. Based on the analysis of antiSMASH outputs, the resulting enzymatic reactions take into account redox cofactors and energy demand. The reactions determined in this way are compatible with the BiGG database, promising a facilitated integration of the biosynthesis pathways encoded in the well-characterised or not BGCs into the GSMNs. Finally, in the case of orphan metabolites, it will be necessary to create both the metabolites and the reactions underpinned by genomic sequences. To maintain optimal traceability, the addition of these entities will be supported by as much metadata as possible such as bibliographic references for the constitution of biosynthetic pathways, InChI-Key, InChI, PubChem identifiers, molecular weights, charges and SMILES until their incorporation into public databases. In addition to the biological relevance and the creation of a model as close to reality as possible, the interest in precisely recreating these pathways lies in the possibility of subsequently analysing in greater detail the distribution and flux reallocation from the precursors of all the targeted metabolites.

The last point concerns the interoperability between specialised metabolic databases and reconstruction databases. The LOTUS database [43] (*i.e.* database of specialised metabolites classified by taxonomy) is queried to target specialised metabolism more specifically. Of the 240 hits found for *P. rubens*, only 47 compounds are potentially usable (*i.e.* metabolites mapped based on their InChIKey versus MetaCyc). This low proportion of integrable data is even more marked if we focus on the Natural Product Atlas database [96], a microbial NP resource containing, among others, referenced data for structure, compound names and source organisms. Still, based on the InChIKey string, 102 compounds were reported for *P. rubens*, but only 6-aminopenicillanate was present in MetaCyc (v23.0) as in the network structure. Extending the search to the genus *Penicillium*, 2,159 NPs are reported. However, only 20 of these compounds have an exact match on MetaCyc, and 44 molecules are raised, looking only at the first section of the InChIKey. Among them, 17 are present in *i*Prub22, including the following seven exact matches: citreoisocoumarin, griseophenone C, brevinamide F,

verruculogen, paxilline, 6-aminopenicillanate and $\beta$-10-hydroxy-12-demethyl-11,12-dehydro-paspaline. These 17 metabolites are involved in 6 consumption reactions, 8 production reactions and 1 reversible transport reaction. These 15 reactions were introduced in the GSMN from either previous networks (*i.e.* 3 from *i*AL1006 and 6 from Prubens), gap-filling (*i.e.* 4 reactions) or functional annotation (*i.e.* 2 reactions). Once again, this reflects the limited compatibility and communication between databases for specific data integration since most of the specialised metabolism in the strict sense found in *i*Prub22 comes from manual processes.

## Conclusion

This study presented a revised version of the *P. rubens* Wisconsin 54–1255 GSMN, encompassing 5,919 reactions, 5,703 genes and 5,192 unique metabolites. The reconstruction process involved updating functional annotations, establishing orthology links with various organisms, and reconciling data from previous *P. rubens* GSMNs.

Developing a functional GSMN is a complex and time-consuming endeavour that necessitates iterative and cyclical steps. These steps are crucial for enhancing our comprehension of the organism being studied and ultimately improving the accuracy of the models. Indeed, the iterative and cyclical nature of the reconstruction process allows for continuous refinement and enhancement of the GSMN. As more data is incorporated and new insights are gained, the models become increasingly precise and representative of the organism's metabolic behaviour. This iterative approach ensures that the models are continuously improved and aligned with the available knowledge. Thus, we have placed significant emphasis on ensuring interoperability, accessibility, and transparency during the *i*Prub22 generation. Consequently, *i*Prub22 respects the actual convention's standards, and we believe that providing a model easily usable by the community will facilitate advancements in the field and encourage knowledge sharing.

The *i*Prub22 model fulfils the fundamental criteria for metabolic modelling, effectively simulating the growth of *P. rubens* by considering variations in carbon and nitrogen, reflecting environmental changes and diverse nutrient sources encountered by the fungus. Moreover, the *i*Prub22 reconstruction serves as a robust backbone for generating different models and investigating the growth and production of specialised metabolites under diverse environmental conditions. It contributes to our understanding of *P. rubens*' metabolic capabilities and provides a valuable resource for future studies in the field of systems biology. Furthermore, we demonstrate that *i*Prub22 exhibits a predictive capacity for specialised metabolite production on the reference medium (*i.e.* glucose as C-source, ammonia as N-source).

Finally, a particular effort was made to explore the specialised metabolism integrated into this model. Although the generation of more data in recent years has led to an increasing size of networks, the percentage of specialised metabolites in the models remains unchanged. Therefore the effort undertaken to update the reconstruction databases seems linear, but the proportion devoted to specialised metabolites warrants further investigation. Exploring the presence of specialised metabolism further highlights the lack of information within the databases and shared identifiers. Such problems need to be addressed before any use of GSMN at a large scale in NP studies.

## Materials and methods

Reconstruction is performed following the steps recommended in the 2010 Reference Protocol [60]. The *i*Prub22 reconstruction is provided in the SBML community standard format [97] with a quality score of 74% evaluated by MEMOTE (v0.13.0) [54]. To ensure transparency and reproducibility, the reconstruction process and related documents are available in Supporting

Information. As a final step, the model was registered in BioModels [98] and given the identifier MODEL2306150001.

### The primary stages: Generating the draft

**Data origin.**    The genomic data for *P. rubens* Wisconsin 54–1255 (*i.e.* 12,557 protein and gene sequences), as well as the General Feature Format file associated, were downloaded from the Ensembl Fungi browser under accession number GCA_000226395 [13].

**Functional annotation.**    *Penicillium rubens* functional annotation is performed using the Trinotate pipeline (v3.2.1) [31] for its boilerplate SQLite database. Thus, homology analyses are launched with blastx and blastp (v2.5.0) with an e-value cut-off of 0.001 against the UniProt database (download on 27 June 2020). The best hit of a sequence is considered its annotation. Search for protein domain is carried out by HMMER (v3.3) [99] against the PFAM database (download on 27 June 2020) with an e-value cut-off of 0.001 and a threefold in the per-domain output of 0.01. SignalP (v4.1g) [100] and TMHMM (v2.0c) [101] are used to search for peptide signals and transmembrane domains, respectively, with their default settings. Protein subcellular localization is determined using DeepLoc (v1.0) [102].

Data that are mandatory for the GSMN reconstruction, Gene Ontology Terms (GOTs) [27,28] and EC numbers are captured respectively using an internal Trinotate script and by querying the KEGG database [29] using its own orthology identifiers (KO). The resulting KO list is enriched by a *P. rubens* proteome analysis performed on the KAAS webserver (Automatic Annotation Server v2.1) [30] against a dataset composed of 367,616 genomic sequences exclusively selected from 37 fungi kingdoms (Best Bidirectional Hit-method and other settings by default–July 2020) and by the exploitation of pre-existing *P. rubens* data [13] on KEGG (Genome information accession–T01091). EC-numbers are then retrieved from these three selected KO sets using KEGG-API REST (REpresentational State Transfer) application programming interface in July 2020.

Annotations combined with the information contained in the *P. rubens* General Feature Format are aggregated to generate a GenBank file using the script: https://github.com/ArnaudBelcour/gbk_from_gff. This file was then used as input to the PathoLogic software from the Pathway Tools suite (v23.0 default settings) [103]. This qualitative metabolic model reconstruction module allows the inference of the reactome from the annotated genome. The database containing the information from the annotation was then exported in attribute-value flat files, which were necessary for further analysis in the AuReMe workspace [63] (see Data consensus and integration for further details).

**Orthology completion.**    Using OrthoFinder (v2.3.12) [32] and Blastp (v.2.5.0), Reciprocal Best Normalized Hit (RBNH) strategy was applied to identify functionally identical genes. Blastp was parametrised with an e-value of 0.001 and performed against seven selected templates: *Arabidopsis thaliana* [33], *Aspergillus nidulans* [34,35], *Aspergillus niger* [34,35], *Neurospora crassa* [36], *Penicillium* species complex [34,35], *Saccharina japonica* [37] and *Schizosaccharomyces pombe* [34,35]. Upstream, data from the different selected templates were filtered to keep only the sequences present in the associated GSMN models.

**External sources.**    The reconstruction is enriched and completed by external data sources. First, data from the Pathway/Genome Database Concepts **PchCyc** (PGDB belonging to Tier 2) available on BioCyc are searched from the special SmartTables [38] provided. Second, compliant information present in the first [26] and latest [39] versions of the network are extracted and parsed to be added to the draft. Only those reactions supported by genomic information that either has a MetaCyc compatible reaction ID or reactions for which all reactants and

products have a MetaCyc ID (in which case the original reaction ID is retained) are incorporated into the draft.

**Data consensus and integration.** AuReMe (AUtomatic REconstruction of MEtabolic models–v2.4) dedicated to GSMNs reconstruction [63] was used to reconstruct the *P. rubens* GSMN. This ToolBox encapsulates the various programs needed to create a high-quality network and maintain a consistent steps record. Results obtained from PathwaysTools and OrthoFinder are injected into AuReMe to generate the intermediate subnetworks resulting from the functional annotation and the orthology search, respectively. Reconstruction was based on the data present in MetaCyc (v23.0) [104] As data templates for orthology refer to other databases like KEGG [29] or BiGG [105], a mapping operation using MetaNetX dictionary [53] (*i.e.* intrinsic to AuReMe MNXref version 2018/09/14) was performed to obtain identifiers compliant with MetaCyc. These data are then merged and completed with the information from external sources. Gene annotations are evaluated and compared using the Fungi-Fun [106] web server (v2.2.8 Beta). The resulting draft is then analysed both qualitatively using topological analysis and quantitatively using constraint-based analysis, refined by manual curation and enriched with data extracted from the literature (See the following section).

The majority of the Figures are drawn using R (v3.5.1) and the following packages: ggplot2 (v3.3.5) [107], hrbrthemes (v0.8.0), ggpubr (v0.4.0) for visualisation and plyr (v1.8.6), dplyr (v1.0.7) and forecast (v0.5.1) for data formatting. Comparisons are made using the Venns diagram (http://bioinformatics.psb.ugent.be/webtools/Venn) for the simplest cases and upsetplot with UpSetR (v1.4.0) [108] for the most complex cases. For the visualisation of the network with KEGG maps, KEGG mapper was used [109].

## The final stages: from draft to high GSMN quality

**Potential producibility–Topological analyses and manual curation.** The reconstruction is refined by examining the presence of known compounds (*i.e.* called targets) in *P. rubens*. The metabolites list to be verified is first established according to data described in the literature, molecules isolated in the laboratory and results identified by antiSMASH/fungiSMASH [42]. This list is enriched by metabolites present in the LOTUS database (**https://lotus.naturalproducts.net**) [43] to focus more precisely on the specialised metabolism. Lastly, the metabolites present in the draft are filtered according to the annotations containing the term "secondary" in the MetaCyc ontologies.

Then, the potential topological producibility tests were carried out using the MeneTools suite (v3.2.0) [110]. It was necessary to define a list of initiating compounds to perform topological analyses (*i.e.* starting point called a seed). All metabolites found in the extracellular compartment were considered as seeds to model the nutritional environment impact (*i.e.* system boundaries). Although detailed intracellular compartmentation was not modelled within *i*Prub22, particular attention was given to the exchange between intracellular and extracellular environments due to the nutritional regime of the fungus (*i.e.* extracellular absorption). The added transport reactions were mainly the result of the information concatenation from *i*AL1006, PchCyc, and some literature. GPR associations were checked and cleaned up according to the annotation provided by DeepLoc [102]. The subcellular assignment of each gene product is available within the SBML in the notes tag of the gene entities. For modelling purposes, artificial exchange reactions (*i.e.* uptake for the input of a compound and production for the output) were added to the reconstruction. They were defined according to the metabolites present in the extracellular medium, which possessed a transport reaction towards the intracellular compartment. The seeds, therefore, corresponded to all the metabolites that have an uptake reaction (S4 File). Artificial genes of form t001 to t208 (*i.e.* transport), d001 to d037

(*i.e.* demand), u001 to u185 (*i.e.* uptake), sk001 (*i.e.* sink) and p001 to p042 (*i.e.* production) were assigned for each of them to discretise specific reactions more efficiently than those coming from a simple gap-filling.

Network gap-filling was performed using the MENECO tool (v2.0.0) [111] This topological gap-filling tool seeks to make accessible from a given list of seeds a set of targets. As the gap-filling process is based on heuristics, it was decided to use filtered repair networks corresponding to MetaCyc subsets. They were built and queried successively, independently, and then jointly until the entire MetaCyc database (v23.0) was used. The MetaCyc subsets were divided into two main categories. On the one hand, the subset was composed of reactions reported in fungal reconstructions (*i.e.* using the following 3 PGDBs available on MetaCyc–*Candida albicans* SC5314, *Exophiala dermatitidis* NIH/UT8656 and *Saccharomyces cerevisiae* S288c –and a fungi meta-network [44]). On the other hand, the subset was composed of reactions for which no genomic data is associated (*i.e.* spontaneous reactions or unknown enzymes). Gap-filling processes are performed iteratively from the targets with the highest confidence on the most relevant repair networks. The resulting reactions are manually verified and incorporated into the network (reactions and their sources are available in the S5 File).

For instance, this process was used to enrich the reconstruction with spontaneous reactions specifically. As Palsson *et al.* [60] recommended, only reactions for which at least one of the entities (*i.e.* reactant or product) was already present in the draft were added to the reconstruction. As it was done for previous specific reactions, an artificial gene of the form s001 to s086 is assigned for each of them.

Objects such as metabolites or reactions are stored, listed, filtered and sorted using Smart-Tables available on MetaCyc [38]. Reconstruction curation and debugging are manually assisted by the network visualisation *via* ModelExplorer (v2.1) [51].

**From reconstruction to models.** The transformation of the reconstruction into a usable model was carried out using MATLAB (v2018b) and diverse functions of the COBRA Toolbox (v3.0) [112].

As with reconstruction, model refinement requires several iterative correction and adjustment processes to optimise the model's performance. Moreover, these crucial steps improve model predictions' alignment with experimental observations, increasing its fidelity to *P. rubens* biological system.

The first point we addressed concerned the reversibility of reactions. To ensure accuracy and reliability, we meticulously updated the reversibility of reactions following the evolving information and reactions with indeterminate directionality were appropriately blocked. The second improvement concerns the naming redundancy issues related to compound class encompassing multiple specific metabolites. For example, if a compound class like D-galactopyranose includes ALPHA-D-GALACTOSE and GALACTOSE, and there is an interconversion reaction between their alpha and beta forms, we prioritised the use of ALPHA-D-GALACTOSE to replace D-galactopyranose. Then, we focused on reaction redundancy related to identifier inconsistency (*e.g.* a reaction with MetaCyc identifier and a redundant reaction with "homemade" identifier). In such a case, we retained the more balanced reaction. Additionally, we ensured that the GPRs were identical or that all the genes from one reaction were included in the other. Discarded reactions were thus closed. The third aspect covered imbalanced reactions. The checkMassChargeBalance() function was employed to check mass and charge balance for every reaction. As a result, a large majority of reactions detected to be unbalanced were blocked.

Consequently, to guarantee that the simulations run as expected, we have established three models: the default, the open and the closed model. The default model represents the minimum uptakes necessary for biomass production (details provided in the next section). The

open model allows for all uptakes with an upper bound of 10 mmol.gDW$^{-1}$.h$^{-1}$, while the closed model restricts all uptakes, disallowing any external metabolites from entering the system.

Complementarily, cycleFreeFlux() and findMassLeaksAndSiphons() functions were used on the closed model. Theoretically, without nutrient input, the results of these functions must be null or at least minimal. The first function allows the identification of thermodynamically infeasible cycles within the metabolic network to ensure a more realistic representation of cellular metabolism. The second function identifies any potential mass leaks (*i.e.* molecular species can be generated from nothing) or siphons (*i.e.* molecular species consumed without yielding anything) in the model. Furthermore, based on the work of Fritzemeier et al [87] and their recommendation regarding energy-generating cycles, we verified by adding an energy-dissipating reaction for 13 compounds (*i.e.* ATP, CTP, GTP, UTP, ITP, NADH, NADPH, FADH2, FMNH2, Ubiquinone-8, acetyl-CoA, glutamate, proton) that no flux was generated for these reactions when the model is closed.

Lastly, after addressing all these aspects, we proceeded to evaluate the predictive performance of *i*Prub22 using analyses related to constraint-based modelling, namely FBA and FVA. Additionally, Fluxer [113], a web application combining FBA and graph theory to offer an interactive graph visualisation, was also used to help us in the flux model refinement. These analyses allowed us to assess the metabolic capabilities of the model, predict optimal flux distributions under different conditions, and explore the range of possible flux values for each reaction.

**Model functional capabilities: biomass and specialised metabolites production.** The 2013 network, *i*AL1006, proposes a biomass reaction adapted from the organism's behaviour [26]. Thus, ten subsystems that model the production of amino acids, cell wall components, cofactors, DNA, RNA, fatty acids, phospholipids, glycerides, sterol esters, and some generic lipids define the biomass reaction (S5 File). The complete adaptation of this physiologically relevant objective function to the data on MetaCyc was made impossible due to the lack of clear mapping between the identifiers and the use of highly generic terms. To test the validation of our proposed model, the biomass reaction was identical to that present in the 2018 network, Prubens [39].

Then, for biomass production analyses, *i*Prub22 was subjected to growth simulations on different media compositions (S6 File), including modifications of the carbon source (35 simulations), nitrogen source (35 simulations), and the use of amino acids as combined carbon and nitrogen sources (21 simulations). Considering that the C-source required for respiration is significantly higher than the N-source, C-source was fixed at 15 mmol.gDW$^{-1}$.h$^{-1}$, while the N-source was set at 5 mmol.gDW$^{-1}$.h$^{-1}$. When simulating the utilisation of amino acids as both carbon and nitrogen sources, their uptake bounds were also set to 15 mmol.gDW$^{-1}$.h$^{-1}$. The uptakes of thiamine (Uptake_171), ferrous ion (Uptake_062), sulphur (Uptake_169), riboflavin (Uptake_157), and phosphate (Uptake_146) were set to a value of 10, while the uptake of oxygen (Uptake_136) remained unrestricted. The defined constraints allowed us to systematically evaluate and analyse the models' behaviour and metabolic capabilities across diverse growth conditions.

For specialised metabolite production analyses, FVA was performed with the biomass production maximised arbitrarily at 80% and reference constraints were applied (*i.e.* glucose as C-source, ammonia as N-source, and uptake openings at the same rates mentioned earlier). Given that the biosynthesis pathways of many specialised metabolites involve a flavoprotein containing both FMN and FAD, the corresponding uptake was open (Red-NADPH-Hemoprotein-Reductases—Uptake_155). It was necessary to unblock the decanoate (Uptake_044)

and phenoxyacetate (Uptake_144) uptakes for penicillin K and V production, respectively. These 3 uptakes were also opened arbitrarily at 10 mmol.gDW$^{-1}$.h$^{-1}$.

Finally, in adherence to the MIASE (Minimum Information About a Simulation Experiment) recommendations [48] to maintain a record of the modifications made to the reconstruction and to ensure comprehensive documentation of the model modifications, the details of the changes operated on the model were organised in a LiveScripts file from MATLAB (S7 File) and the simulation results (S6 File) are available in the Supporting Information.

**SBML format.** The network resulting from AuReMe was modified following the current writing conventions [25,97] to ensure the sustainability and viability of the model. To comply with these requirements (SBML Level 3, FBC2 [59,74]), a format update was performed by relying on the official recommendations available on the COMBINE resource (**https://co. mbine.org** –community for the coordination of modelling standards in biology). The interoperability and enrichment of the identifiers of the various entities of the model were achieved using MetaNetX (v4.1) [53]. These annotations were encoded within SBML according to the standards of the MIRIAM resource [114] (Minimum Information Requirements in Biochemical Model Annotation). Each entity of the network was also associated with an SBO term [55,56] (Systems Biology Ontology–a nested classification scheme for grouping model components). The conversion to FBC2 [74] was performed *via* the script available on the LibSBML application programming interface [115]. These format modifications were then checked and validated *via* the scripts made available to the community on this same application (SBML Validator–testing the syntax and internal consistency of an SBML file). Finally, the model is tested by MEMOTE (MEtabolic MOdel Tests–v0.13.0) [54] and its report is available in Supporting Information (S8 File).

To conclude, as we have decided not to remove any information from the reconstruction, the model that we propose on BioModels (MODEL2306150001) with its FROG report corresponds to a parametrisation of the reconstruction where potentially incorrect reactions are closed and where the minimal uptakes that ensure both the production of biomass and specialised metabolites are open.

## Supporting information

**S1 File. List of additional tables and figures. S1 Table:** Functional annotation details; **S2 Table:** Features and results of the seven templates used for the reconstruction of the intermediate subnetwork from the orthology search; **S3 Table:** List of reactions involved in the synthesis of roquefortines and meleagrin added to the reconstruction (MetaCyc identifiers); **S4 Table:** Lists of reactions involved in penicillins biosynthesis; **S5 Table:** Identification comparison of BGCs between antiSMASH and fungiSMASH; **S1 Fig:** Overview of *Penicillium rubens* functional annotation; **S2 Fig:** Overview of *Penicillium rubens* reactions performed with KEGG Mapper; **S3 Fig:** Bipartite graph representing the seven templates networks' topology for the orthology subnetwork generation; **S4 Fig:** Sequences number distribution per orthogroups according to species detected with OrthoFinder; **S5 Fig:** Visualisation of orthologous genes detected by OrthoFinder and inference of their reactions to the orthology subnetwork; **S6 Fig:** Sankey plot of the data selection from external sources; **S7 Fig:** Scatter plot of reconstruction sources complementarity; **S8 Fig:** Classification of genes and reactions according to their source of integration in the draft; **S9 Fig:** Classification of genes and reactions according to their associated Enzyme Commission number; **S10 Fig:** Origin of metabolites in *i*Prub22 according to reaction reconstruction sources; **S11 Fig:** Sources of transport and exchange reactions added to the reconstruction; **S12 Fig:** Distribution of the 510 reactions added to the reconstruction during the gap-filling steps; **S13 Fig:** Annotations enrichment of the 3,771

genes added to the *Penicillium rubens* GSMN reconstruction (Results from FungiFun).
(PDF)

**S2 File. Orthology results.** This workbook presents an overview of the OrthoFinder results, including selected protein sequences, the number of orthologous sequences, orthogroups between templates, and species-specific statistics.
(XLSX)

**S3 File. Metabolites belonging to *Penicillium rubens*.** This workbook contains a traceability record of the three target lists used for reconstruction refinement, supplemented by a list of orphan metabolites.
(XLSX)

**S4 File. List of transport and exchange reactions added to the reconstruction.** This workbook comprises a comprehensive list of transport and exchange reactions incorporated into the model. It includes transport reactions initially present in the draft, those added with or without gene support, and the necessary exchange reactions for model simulation (Uptake, Demand, Sink, and Production).
(XLSX)

**S5 File. List of reactions from external sources and gap-filling used for *i*Prub22 reconstruction.** This workbook contains reactions from external sources (with or without MetaCyc identifier), spontaneous reactions, the four sets of reactions from gap-filling, and the three reactions required for biomass synthesis (*i.e.* biomass formulation, transport, and exchange).
(XLSX)

**S6 File. List of modified reaction bounds and model simulations.** This workbook includes information on reaction bounds adjusted during the model generation and presents simulations of *P. rubens* growth on different media using FBA with diversified constraints. It also explores the potential production of specialised metabolites using FVA on the reference medium.
(XLSX)

**S7 File. Features of reconstruction and model.** This Livescript MATLAB showcases the characteristics of the reconstruction process and outlines the modifications required for generating the model. It provides a detailed analysis of the reconstruction and model development.
(GZ)

**S8 File. MEMOTE report of *i*Prub22 reconstruction.** This file is an informative report that details the MEMOTE test suite results conducted on the *i*Prub22 reconstruction.
(HTML)

## Acknowledgments

The authors gratefully acknowledge the financial support of the French National Research Agency (ANR), project number ANR-18-CE43-0013. We gratefully thank Erwan Corre (Sorbonne Université, CNRS, FR2424, ABiMS, Station Biologique de Roscoff, Roscoff, France) and David Touboul (Université Paris-Saclay, CNRS, Institut de Chimie Des Substances Naturelles, UPR 2301, Gif-Sur-Yvette, France) for their valuable advice. We also gratefully acknowledge Jeanne Got and Anne Siegel from the Institute for Research in IT and Random Systems (IRISA) for their helpful advice and for generously sharing their data. Furthermore, we express our gratitude to the Roscoff Bioinformatics platform ABiMS (http://abims.sb-roscoff.fr),

which is part of the Institut Français de Bioinformatique (ANR-11-INBS-0013) and BioGenouest network, for providing computing and storage resources.

## Author Contributions

**Conceptualization:** Abdelhalim Larhlimi, Samuel Bertrand.

**Data curation:** Delphine Nègre.

**Formal analysis:** Delphine Nègre.

**Funding acquisition:** Samuel Bertrand.

**Investigation:** Samuel Bertrand.

**Methodology:** Delphine Nègre, Abdelhalim Larhlimi, Samuel Bertrand.

**Project administration:** Abdelhalim Larhlimi, Samuel Bertrand.

**Supervision:** Abdelhalim Larhlimi, Samuel Bertrand.

**Validation:** Abdelhalim Larhlimi.

**Writing – original draft:** Delphine Nègre.

**Writing – review & editing:** Delphine Nègre, Abdelhalim Larhlimi, Samuel Bertrand.

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
