## [Decision Letter · Decision Letter 0]

3 May 2023

PONE-D-23-10574Reconciliation and evolution of Penicillium rubens Genome-Scale Metabolic Networks – What about specialised metabolism?PLOS ONE

Dear Dr. BERTRAND,

Thank you for submitting your manuscript to PLOS ONE. After careful consideration, we feel that it has merit but does not fully meet PLOS ONE’s publication criteria as it currently stands. Therefore, we invite you to submit a revised version of the manuscript that addresses the points raised during the review process.

We look forward to receiving your revised manuscript.

Kind regards,

Bhanwar Lal Puniya, Ph.D.

Academic Editor

PLOS ONE

Journal Requirements:

Additional Editor Comments:

When revising the manuscript, the authors are advised to thoroughly consider the recommendations suggested by Reviewer #2 and also take into account the feedback provided by Reviewer #1 regarding enhancing the results of specialized metabolism. The authors should also make all the minor changes suggested by both reviewers with any other necessary linguistic adjustments. 

Reviewers' comments:

Reviewer's Responses to Questions

**Comments to the Author**

1. Is the manuscript technically sound, and do the data support the conclusions?

Reviewer #1: Yes

Reviewer #2: Yes

2. Has the statistical analysis been performed appropriately and rigorously? 

Reviewer #1: Yes

Reviewer #2: N/A

3. Have the authors made all data underlying the findings in their manuscript fully available?

Reviewer #1: Yes

Reviewer #2: Yes

4. Is the manuscript presented in an intelligible fashion and written in standard English?

Reviewer #1: Yes

Reviewer #2: Yes

5. Review Comments to the Author

Reviewer #1: Date: April 27, 2023

Title: Reconciliation and evolution of Penicillium rubens Genome-Scale Metabolic Networks – What about specialised metabolism?

Journal: PLOS ONE

In this paper “Reconciliation and evolution of Penicillium rubens Genome-Scale Metabolic Networks – What about specialised metabolism?”, Nègre & Larhlimi et al. have mainly discussed in detail the reconstruction process, starting from draft reconstruction, manual curation, gap-filling and different versioning of GSMM (iPrub22) of Penicillium rubens, strain Wisconsin 54-1255 using various tools and sources. The whole paper is more focused on reconstruction process and providing community a new knowledge base. Although, authors have briefly given emphasis on specialized metabolism but then missed the opportunity to elaborate and demonstrate it with some case studies, by choosing target metabolites from the list that author have provided. This would have made the paper more useful and brought the novelty.

I am recommending this paper for major revision.

My suggestion for authors is to investigate specialized metabolism and select target metabolites, demonstrate it with 3-4 examples, show the simulation results, along with the complete metabolic pathways for the metabolites and validate your results with literature data if infrastructure for experimental validation is lacking.

Please find below some of the other minor comments below.

Abstract:

Line 15: remain silent in laboratory => you can use: are not exploited.

Line 18-19: … clever alternative?  possible alternative

Line 22-27: In parallel ………specialized metabolites. Should be part of results section.

You have reconstructed the model which I agree is an important result itself, I am wondering you started the abstract discussing about the metabolites synthesized by biosynthetic gene clusters that have not been exploited which should have been the novelty of this study, and then you have given more stress on reconstruction process.

Line 39,40,41: molecules – rather than use compounds/metabolites

Line 49: relaunch  study or investigate.

Line 84-85: properties of metabolism? organisms can be described…  metabolic properties of an organisms can be described…

Line 91-92: enhancing our metabolism understanding at the system level [17].  enhancing our understanding of metabolism at the system level [17].

Line 96: whose non-exhaustive? list is presented…

Line 104: (e.g., knowledge snapshot)?

Line 106: pressing  important.

Line 130-131: Fig 1 in S2 file Veen  Venn

Line 133-134: compared to the only benefit? of Trinotate

Line 157-158: Fig 4 A-B in the S2 file  center align x-axis title. The Fig 4 A can be made better.

Fig 9, 10,11 in the S2 file, center align x-axis & y-axis title. Fig 8 in the S2 file, center align x-axis.

Line 253: searchable model?

Line 255: These Target compounds presence in the ….  These Target compounds/metabolites present in the …

Line 260: Title: Targets selection  Target selection

Line 279: a list of 47 compounds lacking…. S4 file sheet Orphans metabolites  46 metabolites, header is not a metabolite.

Line 284-285: targets1 … 243 targets  237 mentioned in S4 file sheet Targets1. Check discrepancy between table S4 & Table 1

Line 296: Targets2 …35  47 mentioned in S4 file sheet Targets2. Check discrepancy between table S4 & Table 1.

Align the titles for x-axis and y-axis for supp. And main figures in the paper to center

add readme for tables in s3 file.

Reviewer #2: This manuscript introduces a new semi-automatically built and manually curated genome-scale metabolic model for the organism P. rubens strain Wisconsin 54-1255 based on a newly annotated genome. The model is comparably large and follows common standards. The result is a well-standardised and reusable model with great potential for other researchers in systems biology. The authors should only improve a few aspects before publication.

1) Major Points

The SBML model

The model is entirely valid, and its MEMOTE score is quite sound. Many sub-categories even reach the maximum of 100%. However, the 68% still have room for improvement. In particular, links to the BiGG Models database are scarce. Other works use the tool ModelPolisher (https://github.com/draeger-lab/ModelPolisher) to increase the model score. The authors could give it a try.

Also, the authors should upload the model to the BioModels database instead of providing it as a supplement for multiple reasons:

1. It will make their model more easily findable because modellers can search for models in one central place and find the model with the link to this publication.

2. Model revisions can be created, which is different for supplements. In particular, if, despite careful reconstruction, inaccuracies are found in the model later on, this versioning will be beneficial.

3. The BioModels team has professional curators who permanently update and improve models and ensure their reusability.

When doing so, it would be ideal if the authors could include a FROG analysis in their model and wrap it in an OMEX file (see https://www.ebi.ac.uk/biomodels/curation/fbc for details).

As a minor note: The model’s annotation should link the NCBI taxon identifier using the hasTaxon qualifier instead of “is”.

2) Minor Points

References

* Please support the development of SBML by citing the specific SBML specification documents in addition to the review by Keating et al. (2020). In this case, references are necessary to the specification for SBML Level 3 Version 1 (https://doi.org/10.1515/jib-2017-0080) and the FBC package version 2 (https://doi.org/10.1515/jib-2017-0082).

* The paper by Courtot et al. (2011) is a reference that describes all ontologies with relevance for SBML in one overview (see https://doi.org/10.1038/msb.2011.77) and is typically cited to refer to SBO and so forth.

* The newer article https://doi.org/10.1093/nar/gkz1054 supersedes reference number 100 and should be cited instead.

Style and other remarks

* Please avoid starting sentences with lowercase letters, as this is done in line 121.

* It is a custom that the words “Level” and “Version” are written with uppercase letters in conjunction with SBML rather than in lowercase.

* The COMBINE website is now reachable via HTTPS from https://co.mbine.org (no longer just HTTP).

6. PLOS authors have the option to publish the peer review history of their article (what does this mean?). If published, this will include your full peer review and any attached files.

Reviewer #1: **Yes: **Ab Rauf Shah

Reviewer #2: No

---

## [Author Response · Author response to Decision Letter 0]

6 Jul 2023

Revised Manuscript and Responses to Reviewers' Comments

Preamble

During the correction process, we obtained complementary results that led us to update some aspects of the manuscript. This yield to an improved model whose simulated growth possess improved sensitivity to nutrient variations. Below is a list of the modifications made:

• Updating and correcting all the numbers in the manuscript

• Addition of two figures containing the results of flux simulations - as asked by the reviewers - on growth (Fig 3) and the production of specific metabolites by P. rubens (Fig 6)

• Addition of a supplementary data file containing the bounds modifications carried out on the model and the results of the various simulations carried out.

• Modifications and reorganisation of the sections "Reconstruction transformation into a constrained model", Connectivity and metabolic coverage", and "Specialised metabolism through versions" (Results section)

• Reorganisation of the "From reconstruction to a model that simulates environmental exchanges" section (Discussion section)

• Update of "From reconstruction to models" and "Model functional capabilities: biomass and specialised metabolites production" (Material & Methods section)

Reviewer #1: Date: April 27, 2023

[REVIEWER COMMENT] Title: Reconciliation and evolution of Penicillium rubens Genome-Scale Metabolic Networks – What about specialised metabolism?

Journal: PLOS ONE

In this paper “Reconciliation and evolution of Penicillium rubens Genome-Scale Metabolic Networks – What about specialised metabolism?”, Nègre & Larhlimi et al. have mainly discussed in detail the reconstruction process, starting from draft reconstruction, manual curation, gap-filling and different versioning of GSMM (iPrub22) of Penicillium rubens, strain Wisconsin 54-1255 using various tools and sources. The whole paper is more focused on reconstruction process and providing community a new knowledge base. Although, authors have briefly given emphasis on specialized metabolism but then missed the opportunity to elaborate and demonstrate it with some case studies, by choosing target metabolites from the list that author have provided. This would have made the paper more useful and brought the novelty.

[ANSWER] First of all, we would like to thank you for reviewing our manuscript, "Reconciliation and evolution of Penicillium rubens Genome-Scale Metabolic Networks – What about specialised metabolism?" We appreciate your thoughtful comments and the time you have taken to provide us with constructive feedback. We understand your point about the need to provide more emphasis on specialised metabolism and hope that the answers we have provided in this document will explain and clarify the choices we have made. Thus, we have addressed each of your comments and provided our responses below.

[REVIEWER COMMENT] I am recommending this paper for major revision.

My suggestion for authors is to investigate specialized metabolism and select target metabolites, demonstrate it with 3-4 examples, show the simulation results, along with the complete metabolic pathways for the metabolites and validate your results with literature data if infrastructure for experimental validation is lacking.

While we understand the value of such an investigation, we believe that incorporating it in the current manuscript would significantly expand the scope and density of the work. However, to address the concern about specialised metabolites, we have included a column in our supplementary data (S3 file, formerly S4 file) indicating the topologically producible target metabolites. Furthermore, we have focused on ensuring the accurate simulation of growth prerequisites by the model, as demonstrated in Fig 3. Regarding specialised metabolite production, we have included some results in the manuscript (Fig 6). We acknowledge that these results need to be validated through experimental means, as suggested by the reviewer. We hope that by providing this foundation and indicating the producibility of specialised metabolites, our work lays the groundwork for future investigations and experimental validations.

[REVIEWER COMMENT] Please find below some of the other minor comments below.

Abstract:

Line 15: remain silent in laboratory => you can use: are not exploited.

Line 18-19: … clever alternative?  possible alternative

Line 22-27: In parallel ………specialized metabolites. Should be part of results section.

You have reconstructed the model which I agree is an important result itself, I am wondering you started the abstract discussing about the metabolites synthesized by biosynthetic gene clusters that have not been exploited which should have been the novelty of this study, and then you have given more stress on reconstruction process.

[ANSWER] Thank you for your feedback on our abstract. Following your comments, we have restructured our abstract to make it more concise, accessible and, more accurate. We address each of your points in detail below.

First, we have changed "remain silent in laboratory" to " does not necessarily guarantee their expression under laboratory conditions" to emphasise the current challenges associated with exploiting these metabolites in the experimental field. For the second point, we agree that the word "clever" would be too impactful. We have therefore qualified this expression by " represents a promising approach".

Then, "In parallel, (1) an updated functional annotation of the P. rubens genome was carried out and supplemented by (2) an orthology search with different GSMNs templates. This first draft was enriched (3) by integrating data from P. rubens previous GSMN reconstructions and complemented (4) by manual curation steps targeting basal and specialised metabolites" may be confusing with results, whereas it represents the protocol we followed for the reconstruction. For the sake of clarity, we have reworded this sentence as follows: "Our reconstruction, iPrub22, adheres to current convention standards and quality criteria, incorporating updated functional annotations, orthology searches with different GSMN templates, data from previous reconstructions, and manual curation steps targeting basal and specialised metabolites."

Finally, given the major corrections that you propose to improve our manuscript, we understand your question concerning the beginning of our abstract. The exploitation of specialised metabolites of filamentous fungi is indeed the final objective and the reason for the proposal of our new reconstruction. Nevertheless, the reconstruction of a metabolic network and the analysis of the resulting constraint models are, from our point of view, two disciplinary fields that are certainly indissociable but also distinct. Although our manuscript provides preliminary insights into the presence and topological producibility and flux production of specialised metabolites, we have chosen to retain the concept of specialised metabolism of filamentous fungi as our catchphrase. We hope that the simplifications we made best reflect the content of the paper we are presenting, and we thank you again for the time you took to suggest improvements to the abstract of our manuscript.

[REVIEWER COMMENT] Line 39,40,41: molecules – rather than use compounds/metabolites

[ANSWER] We thank you for your attention to our manuscript to make it more rigorous in these definitions. The term "molecule" is generic, and the word "metabolites" refers directly to the products of metabolism. So, following your recommendations, we have changed two occurrences of the word "molecules" to "metabolites" to gain precision. However, we kept the first occurrence to avoid redundancy and underline, once again, that we are specifically interested in living organism products.

[REVIEWER COMMENT] Line 49: relaunch  study or investigate.

[ANSWER] Thank you for your suggestion of simplification. However, with this sentence, we aim to emphasise the need to revitalise a research area that is losing momentum. To clarify our thinking, we have slightly modified our sentence by adding some contextual words and replacing the verb "relaunch" with "reinvigorate".

[REVIEWER COMMENT] Line 84-85: properties of metabolism? organisms can be described…  metabolic properties of an organisms can be described…

[ANSWER] We thank you for your careful reading. We have followed your recommendations for the following two comments.

[REVIEWER COMMENT] Line 91-92: enhancing our metabolism understanding at the system level [17].  enhancing our understanding of metabolism at the system level [17].

[ANSWER] Done.

[REVIEWER COMMENT] Line 96: whose non-exhaustive? list is presented…

[ANSWER] With this comment, we understand that our sentence is not clear and that we do not convey the desired idea. For sake of clarity, we reversed the order of the propositions in this sentence to emphasise that, apart from the networks of model organisms (which are the result of multi-collaborative work over the years), there is a lack of this type of work on other organisms.

[REVIEWER COMMENT] Line 104: (e.g., knowledge snapshot)?

[ANSWER] Thank you for bringing this to our attention. By "knowledge snapshot," we mean a specific moment in time when the reconstruction of the metabolic network was created, which represents the best understanding of the organism's metabolism based on the available knowledge and data at that time. We have revised the sentence to clarify this point.

[REVIEWER COMMENT] Line 106: pressing  important.

[ANSWER] We acknowledge that the term "pressing" might suggest a stronger sense of urgency than intended, and we appreciate your suggestion to use the word "important" instead. However, we have decided to keep the original wording as we believe it effectively conveys the urgency and criticality of maintaining up-to-date and standardised models, particularly given the constant and rapid evolution of data. We hope that this clarifies our reasoning behind the choice of language in the manuscript.

[REVIEWER COMMENT] Line 130-131: Fig 1 in S2 file Veen  Venn

[ANSWER] Thank you for your careful review of our additional data. We have corrected the grammatical error concerning the word "Venn" in the caption of Figure S1 in the S2 file.

[REVIEWER COMMENT] Line 133-134: compared to the only benefit? of Trinotate

[ANSWER] Thank you for pointing out the lack of clarity in our wording. We hope that the rephrasing will make it more readable.

[REVIEWER COMMENT] Line 157-158: Fig 4 A-B in the S2 file  center align x-axis title. The Fig 4 A can be made better.

Fig 9, 10,11 in the S2 file, center align x-axis & y-axis title. Fig 8 in the S2 file, center align x-axis.

[ANSWER] Regarding the figures formatting in the S2 file (rename in S1 file), we have made the necessary adjustments, including centring the axis captions where appropriate and standardising the font sizes for consistency. Additionally, we have followed your suggestion to rework Figure S4 into three panels, which has improved its readability. Finally, we have corrected the order of figures S7, S8, S9, and S10 to match the manuscript's original sequence.

[REVIEWER COMMENT] Line 253: searchable model?

[ANSWER] Thank you for your comment. We agree that our vision of reconstruction needs clarification. As we see it, the usefulness of a reconstruction goes beyond the predictive power of the resulting model. For instance, it can also provide valuable information for researchers studying the organism, such as identifying the genomic sequences associated with each enzyme through GPR associations. Therefore, we have added an emphasis to define our concept of a "searchable model" (a knowledge platform comprising accessible resources for a given organism at the instance of reconstruction).

[REVIEWER COMMENT] Line 255: These Target compounds presence in the ….  These Target compounds/metabolites present in the …

[ANSWER] Thank you for bringing to our attention the lack of clarity in our sentence. We have revised it to clarify the two distinct logical aspects. While it is crucial for the target compounds to be present in the network and topologically producible, these factors alone are not sufficient for conducting flux studies but rather constitute a starting point. Therefore, these two points are essential and they have been explored in our reconstruction.

[REVIEWER COMMENT] Line 260: Title: Targets selection  Target selection

[ANSWER] Thank you for bringing the grammatical error in our title to our attention. We have taken your feedback into account and corrected it.

[REVIEWER COMMENT] Line 279: a list of 47 compounds lacking…. S4 file sheet Orphans metabolites  46 metabolites, header is not a metabolite.

[ANSWER] We appreciate the thoroughness with which you reviewed our additional data and we have corrected the incorrect data in Table 1. We correct this error and ensure the accuracy of our findings.

[REVIEWER COMMENT] Line 284-285: targets1 … 243 targets  237 mentioned in S4 file sheet Targets1. Check discrepancy between table S4 & Table 1

Line 296: Targets2 …35  47 mentioned in S4 file sheet Targets2. Check discrepancy between table S4 & Table 1.

[ANSWER] To clarify the points mentioned, we have included a README sheet at the beginning of our Excel workbook titled "S3_file.xlsx" (former S4 file) which provides detailed information on the selection of targets. Here are the specifics 

for the Targets1 and Targets2 sheets:

The "Targets1" sheet contains the first set of target metabolites, which were selected based on information from the literature. These target metabolites belong indifferently to constitutive or specialised metabolism. The selection process involved two noteworthy publications: one that enabled accurate modelling of the sugar pathway in fungi (Aguilar-Pontes et al.) and another that resulted from manual curation of the penicillin biosynthesis pathway (Prauβe et al.). The list of targets was complemented by metabolites associated with biomass function extracted from iAL1006 (Agren et al.) and information extracted from the SMASH tool suite. Please note that, out of the 237 identified metabolites presented, Xanthocillin and Pr-toxin have MetaCyc identifiers but are not associated with any reactions. Hence, within iPrub22, only 235 compounds with a MetaCyc Id are potentially exploitable. Finally, to model the biomass reaction, we have included eight homemade identifiers from iAL1006: AAPOOL, CELLWALL, COF, DNA, RNA, PROTEIN, PLIPIDS, and Biomass produced by reactions r1459, r1455, r1465, r1458, r1457, r1456, r1460, and Biomass_rxn, respectively. These identifiers represent artificial compounds that encompass a combination of metabolites. Although they are not listed in the Targets1 sheet, we have included them in our count of 243 targets because their producibility is crucial for ensuring network functionality.

The "Targets2" sheet consists of metabolites obtained by querying the LOTUS database, a specific Natural Product database. Out of the 240 metabolites associated with P. chrysogenum/P. rubens in this database, only 47 are found in the MetaCyc database on the basis of a complete or incomplete InChIKey match. As we decided to retain only those metabolites that are known to be present in Wisconsin strain 54-1255, 11 of these compounds are excluded from our search set. Finally, as the patulin biosynthetic pathway is incomplete in P. rubens, patulin is also removed from our target set. However, these 12 metabolites are still presented in this Excel sheet to highlight the importance of being cautious and careful when selecting data. In the end, the Targets2 sheet includes 35 metabolites.

[REVIEWER COMMENT] Align the titles for x-axis and y-axis for supp. And main figures in the paper to center

[ANSWER] As previously stated, we have considered the feedback regarding the formatting of the figures included in the S2 file, and we appreciate your input. Nevertheless, since we did not observe any misaligned components in the main figures, we have left them unchanged.

[REVIEWER COMMENT] add readme for tables in s3 file.

[ANSWER] Thank you for recommending the inclusion of README for our additional data. We appreciate your input and want to let you know that we have followed your advice. Specifically, we have added a separate sheet to each of our Excel workbooks (S3 to S6 files) that provides a detailed explanation of the contents. We hope that the README inclusion will strengthen our work and greatly facilitate the understanding and interpretation of our data, allowing for better reproducibility and further analysis. Once again, we appreciate your suggestion and the opportunity to improve our work.

In conclusion, we appreciate the time and effort you invested in reviewing our manuscript and providing us with valuable feedback. While we understand and respect your suggestions for major revisions, we have decided to maintain the current direction of the manuscript. However, we want to assure you that we have taken your comments on the form of the manuscript and the figures seriously and have made the necessary adjustments to improve their clarity and readability. Your attention to detail has helped us enhance our work presentation, and we are grateful for your constructive feedback.

 

Reviewer #2: 

[REVIEWER COMMENT] This manuscript introduces a new semi-automatically built and manually curated genome-scale metabolic model for the organism P. rubens strain Wisconsin 54-1255 based on a newly annotated genome. The model is comparably large and follows common standards. The result is a well-standardised and reusable model with great potential for other researchers in systems biology. The authors should only improve a few aspects before publication.

[ANSWER] We express our sincere appreciation for your careful review of our manuscript and thank you for the valuable comments you provided. Furthermore, we are grateful for your recognition of our efforts to propose a manually curated genome-scale metabolic model for P. rubens strain Wisconsin 54-1255 that follows common standards. Your comments have provided us with valuable insights, and they have helped us identify the areas that require improvement before publication. We are pleased to read that you find our new genome-scale metabolic model for P. rubens strain Wisconsin 54-1255 to be well-standardized and with great potential for other researchers in systems biology. Thank you for your time and valuable insights. Below, you will find our responses to the comments you provided.

[REVIEWER COMMENT] 1) Major Points

The SBML model

The model is entirely valid, and its MEMOTE score is quite sound. Many sub-categories even reach the maximum of 100%. However, the 68% still have room for improvement. In particular, links to the BiGG Models database are scarce. Other works use the tool ModelPolisher (https://github.com/draeger-lab/ModelPolisher) to increase the model score. The authors could give it a try.

[ANSWER] We thank you for underlining the reconstruction quality score and suggesting ways to improve it. We greatly appreciate the opportunity of being able to develop here some technical points that are not often mentioned in the manuscript. We hope that the following information will meet your concerns.

Firstly, we would like to explain the limited annotations of metabolites and reactions from the BiGG database in our reconstruction. Our mapping strategy, aimed at augmenting the reconstruction with annotations, relies on data from MetaNetX. Specifically, from the MetaCyc identifiers (our reference database for the reconstruction), we started to search for all the MetaNetX IDs. Subsequently, from these MetaNetX IDs, we retrieved all associated identifiers from other databases, including those from BiGG. The Venn diagram below represents all metabolites (orange) and reactions (purple) from the BiGG and MetaCyc databases that have a MetaNetX ID in version 4.1. The intersection shows that the entities sharing a MetaNetX ID represent a small proportion of the overall data. Consequently, the limited representation of BiGG IDs in our reconstruction reflects the extent of communication between MetaCyc and BiGG through the MetaNetX IDs at the time of reconstruction.

Thank you for suggesting ModelPolisher for improving annotation interoperability and enriching our model by identifying BiGG. We had considered this possibility, but knowing that our anchor for annotation enrichment is the MetaCyc annotations, this tool did not seem appropriate for our needs. Indeed, the description of this tool on https://github.com/draeger-lab/ModelPolisher indicates that "ModelPolisher accesses the BiGG Models knowledgebase to annotate and autocomplete SBML models. Thereby, the program mainly relies on BiGG identifiers for model components". Thus ModelPolisher allows reconstruction to be enriched from pre-existing BiGG identifiers in the reconstruction model, which is not our case.

Now, regarding the MEMOTE score, we fully agree that there is still room for improvement. That is why we are providing an actual screenshot below detailing its calculation (i.e. during the correction process, improvements were made to the model and are described at the beginning of this document). We have observed that improving the BiGG annotations would have had a minimal impact on the overall score (although we fully acknowledge that, in reality, their importance is crucial for widespread reconstruction use). Therefore, during the reconstruction, we have chosen to focus on gene and SBO term annotations since, after the mapping process, our annotations for reactions and metabolites were above 80% and seemed satisfactory to us.

Moreover, without delving into the details of the MEMOTE score calculation, we would like to take advantage of your comment to address some clarifications regarding the results displayed in the report's different sections.

• Consistency

o Stoichiometric Consistency 0%

Stoichiometric consistency is assessed through a binary scoring approach, where a score of 0 is assigned as soon as a metabolite is detected in either the unconserved metabolites or inconsistent minimal net stoichiometry section. As we have not curated the data extracted from MetaCyc, the information presented in the "Consistency" section accurately reflects the inherent characteristics of the elements derived from this database.

• Annotation - Metabolites

o Uniform Metabolite Identifier Namespace 84.4% 

The report specifies that "852 metabolite identifiers (15.59%) deviate from the largest found namespace (BioCyc)". After removing the compartment information from these identifiers, 720 unique metabolites remain, of which 645 map to MetaCyc. The last 75 are either id from the biomass function, compounds added from iAL006 or obsolete terms.

• Annotation - Reactions

o Uniform Reaction Identifier Namespace 87.5%

The report specifies that "758 reaction identifiers (12.81%) deviate from the largest found namespace (BioCyc)". These identified reactions pertain to the modelling reactions (noted Uptake_\\d{3}, Production_\\d{3}, Transport_\\d{3}, Demand\\d{3}, Sink\\d{3}) or to reactions coming from iAL1006 (r\\d{4}).

• Annotation - Genes

o Total Genes 6,171 and Gene Annotations Per Database at 92,4%.

In our reconstruction, to facilitate the exploration of the network, we have associated artificial genes with the modelling and spontaneous reactions, in particular, to differentiate them from the gap-filling reactions. More precisely, P. rubens genes follow the pattern Pc\\d{2}\\g\\d{5}, while artificial genes are denoted as [s|u|p|d|t|sk]\\d{3} (representing spontaneous, uptake, production, demand, transport and sink). As a result, the total number of genes is overestimated, leading to underestimated percentages of gene annotation presence in each database.

• Annotation – SBO Terms

o Metabolic Reaction SBO:0000176

The report states that "A total of 91 metabolic reactions (1.70% of all purely metabolic reactions) lack annotation". These 91 reactions correspond to transport reactions, for which we have chosen to associate more specific SBO terms representing their respective categories.

o Transport Reaction SBO:0000185

The report specifies that "A total of 36 metabolic reactions (11.32% of all transport reactions) lack annotation". These 36 reactions refer to demand reactions (Demand_\\d{3}), which are associated with their corresponding SBO term.

o Demand Reaction SBO:0000628

The report specifies that "A total of 4 reactions (9.76% of all demand reactions)". These 4 reactions mentioned are associated with the 'biochemical reaction' SBO term (SBO:0000176).

o Gene SBO:0000243

The report specifies that "A total of 468 genes (7.58% of all genes) lack annotation". These 468 genes correspond to the artificial genes mentioned above and are associated with an SBO linked to an artificial entity ("SBO:0000291: not monitored by MEMOTE).

We hope that this information will provide sufficient and necessary insight to understand both the reconstruction processes we have chosen and their consequential impact on the MEMOTE score. Once again, we would like to emphasise that we are presenting a reconstruction which provides a default model which requires further refinement (as indicated by the specific results of the MEMOTE report). 

[REVIEWER COMMENT] Also, the authors should upload the model to the BioModels database instead of providing it as a supplement for multiple reasons:

1. It will make their model more easily findable because modellers can search for models in one central place and find the model with the link to this publication.

2. Model revisions can be created, which is different for supplements. In particular, if, despite careful reconstruction, inaccuracies are found in the model later on, this versioning will be beneficial.

3. The BioModels team has professional curators who permanently update and improve models and ensure their reusability.

When doing so, it would be ideal if the authors could include a FROG analysis in their model and wrap it in an OMEX file (see https://www.ebi.ac.uk/biomodels/curation/fbc for details).

[ANSWER] We thank you for highlighting the benefits of uploading our model to the BioModels database. We believe that this step will significantly enhance the dissemination and long-term accessibility of our work. We appreciate your insights, and we agree that there are several advantages to making the model more easily accessible through a central repository. Consequently, we are pleased to inform you that iPrub22 was registered in BioModels and given the identifier MODEL2306150001. In line with your suggestion, we will include a FROG analysis in our model and wrap it in an OMEX file. It facilitates the integration of our model with the BioModels database and ensures compliance with the guidelines provided on their website (https://www.ebi.ac.uk/biomodels/curation/fbc).

[REVIEWER COMMENT] As a minor note: The model’s annotation should link the NCBI taxon identifier using the hasTaxon qualifier instead of “is”.

[ANSWER] We greatly appreciate you taking the time to review in detail the model provided. As a result of your feedback, we have made the necessary modifications to the SBML file.

[REVIEWER COMMENT] 2) Minor Points

References

[ANSWER] We thank you for taking the time to provide us with timely suggestions for including additional missing references in our manuscript. We have made the necessary additions and changes, as per your recommendations, which are listed below. Then, to help the reader understand the concept of ontology, we have added to each mention of the term SBO, the article you suggest. Finally, we have updated the reference concerning the BiGG database. Thus, your feedback helps to ensure that our work is up-to-date and accurately reflects the most recent advances in the field.

[REVIEWER COMMENT] * Please support the development of SBML by citing the specific SBML specification documents in addition to the review by Keating et al. (2020). In this case, references are necessary to the specification for SBML Level 3 Version 1 (https://doi.org/10.1515/jib-2017-0080) and the FBC package version 2 (https://doi.org/10.1515/jib-2017-0082).

* The paper by Courtot et al. (2011) is a reference that describes all ontologies with relevance for SBML in one overview (see https://doi.org/10.1038/msb.2011.77) and is typically cited to refer to SBO and so forth.

* The newer article https://doi.org/10.1093/nar/gkz1054 supersedes reference number 100 and should be cited instead.

[ANSWER] modification were performed accordingly.

[REVIEWER COMMENT] Style and other remarks

* Please avoid starting sentences with lowercase letters, as this is done in line 121.

[ANSWER] We want to thank you for your careful reading of our manuscript. Indeed, as the model name iPrub22 begins with a lowercase letter, we have reversed the order of the words on the line you mention, and lines 458 and 1101 are modified to gain precision.

[REVIEWER COMMENT] * It is a custom that the words “Level” and “Version” are written with uppercase letters in conjunction with SBML rather than in lowercase.

[ANSWER] Your attention to detail has helped us improve the accuracy and precision of our work. Following your feedback, we have made corrections to the case of the words "Version" and "Level" on lines 531, 789, 794, and 1262 to ensure consistency throughout the manuscript.

[REVIEWER COMMENT] * The COMBINE website is now reachable via HTTPS from https://co.mbine.org (no longer just HTTP).

[ANSWER] Thank you for pointing out the inaccuracy with our hyperlink. The correction you suggested has made it possible for us to provide a more accurate and up-to-date link.

Once again, we want to express our sincere gratitude for your thoughtful and insightful comments. Your comments have helped us to gain rigour, and to emphasise the importance of spreading our model within the community. Above all, they enabled us to pinpoint and find the aspects that had previously prevented us from obtaining a fully functional model. Thank you once again for your valuable contributions.

---

## [Decision Letter · Decision Letter 1]

25 Jul 2023

Reconciliation and evolution of Penicillium rubens Genome-Scale Metabolic Networks – What about specialised metabolism?

PONE-D-23-10574R1

Dear Dr. BERTRAND,

We’re pleased to inform you that your manuscript has been judged scientifically suitable for publication and will be formally accepted for publication once it meets all outstanding technical requirements.

Kind regards,

Bhanwar Lal Puniya, Ph.D.

Academic Editor

PLOS ONE

Additional Editor Comments (optional):

Reviewers' comments:

Reviewer's Responses to Questions

**Comments to the Author**

1. If the authors have adequately addressed your comments raised in a previous round of review and you feel that this manuscript is now acceptable for publication, you may indicate that here to bypass the “Comments to the Author” section, enter your conflict of interest statement in the “Confidential to Editor” section, and submit your "Accept" recommendation.

Reviewer #1: All comments have been addressed

2. Is the manuscript technically sound, and do the data support the conclusions?

Reviewer #1: Yes

3. Has the statistical analysis been performed appropriately and rigorously? 

Reviewer #1: Yes

4. Have the authors made all data underlying the findings in their manuscript fully available?

Reviewer #1: Yes

5. Is the manuscript presented in an intelligible fashion and written in standard English?

Reviewer #1: Yes

6. Review Comments to the Author

Reviewer #1: After reviewing the revised manuscript, I am pleased to note that the authors have adequately addressed the feedback provided during the initial round of peer review. However, I have a few minor comments for further improvements. Specifically, the figures still exhibit pixelation and do not meet the necessary quality and resolution standards required for publication. It is imperative that the authors make the necessary enhancements to ensure a publication of high quality.

7. PLOS authors have the option to publish the peer review history of their article (what does this mean?). If published, this will include your full peer review and any attached files.

Reviewer #1: No

---

## [Editor Report · Acceptance letter]

21 Aug 2023

PONE-D-23-10574R1 

Reconciliation and evolution of *Penicillium rubens* Genome-Scale Metabolic Networks – What about specialised metabolism? 

Dear Dr. BERTRAND:

I'm pleased to inform you that your manuscript has been deemed suitable for publication in PLOS ONE. Congratulations! Your manuscript is now with our production department. 

Kind regards, 

on behalf of

Dr. Bhanwar Lal Puniya 

Academic Editor

PLOS ONE